# Compact Conformal Subgraphs

**Sreenivas Gollapudi** [1]   **Kostas Kollias** [1]   **Kamesh Munagala** [2]   **Aravindan Vijayaraghavan** [3]

## Abstract

Conformal prediction provides distribution-free uncertainty guarantees, but often yields large prediction sets in structured domains such as routing, planning, or sequential recommendation. We introduce *graph-based conformal compression*, a framework for constructing compact subgraphs that preserve statistical validity while reducing structural complexity. We formulate compression as selecting a smallest subgraph capturing a prescribed fraction of the probability mass, and reduce to a weighted version of densest $k$-subgraphs in hypergraphs, in the regime where the subgraph has a large fraction of edges. We design efficient approximation algorithms that achieve constant factor coverage and size trade-offs. Crucially, we prove that our relaxation satisfies a *monotonicity* property, derived from a connection to parametric minimum cuts, which guarantees the nestedness required for valid conformal guarantees. We finally validate our algorithmic approach via simulations for trip planning and navigation, and compare to natural baselines.

## 1. Introduction

Conformal prediction (Vovk et al., 2005; Shafer & Vovk, 2007; Angelopoulos & Bates, 2021) provides a general framework for attaching statistically valid uncertainty quantification to black-box predictive models. Given a model with predicted outputs $A$ and true outcomes $B$, conformal methods calibrate a distance or nonconformity score $f(A, B)$ on a held-out calibration set and choose a threshold $d$ so that, under exchangeability of samples, $\mathbb{P}(f(A^*, B^*) \leq d) \geq \phi$, where $(A^*, B^*)$ corresponds to a new test pair. The resulting *conformal set* $S(A^*) = \{B : f(A^*, B) \leq d\}$ contains the true outcome $B^*$ with probability at least $\phi$. Conformal prediction thus converts any point predictor $A$ into a distribution-free set predictor $S(A)$ with finite-sample marginal coverage.

As an example, in route planning and navigation systems, a predictive model proposes a route based on observed contextual features. Such features may include the user profile (e.g., commuter, tourist), real-time traffic conditions, weather, time of day, or the nature of the query (such as "fastest", "scenic", or "kid-friendly"). Despite the model's best prediction, a user may choose an alternate path for unmodeled reasons: personal preferences, temporary closures, or new information encountered en route. Consequently, the model output $A_t$ (the suggested path) at a given epoch $t$ may differ from the actual path $B_t$ that the user traverses. Over time, the system accumulates a history of such pairs $\{(A_t, B_t)\}$, where $A_t$ is the model's suggested route and $B_t$ is the corresponding ground-truth route taken by the user.

The conformal prediction framework now works as follows: For a new prediction $A^*$, we wish to construct a set of plausible user routes $S(A^*)$ that is guaranteed (under exchangeability) to contain the eventual ground-truth route $B^*$ with a prescribed probability level $\phi$. Ideally we would also like $S(A^*)$ to be as compact or small as possible.

The route planning task described above is an example of a conformal prediction task over graphs. This framework also captures other settings where realized behaviors diverge from model predictions, for instance in *trip planning*, where a model suggests an itinerary $A_t$ but the user follows $B_t$. Route planning also arises in routing autonomous vehicles, or in shipment logistics, while trip planning is akin to content recommendations. Other applications include settings where the model predicts a sequence of clinical interventions (while physicians or patients may modify the plan), or predicts actions in business or manufacturing processes. These different applications can be modeled using a common graph-based conformal framework.

### 1.1. Graph-based Conformal Compression

In structured prediction tasks such as route planning, itinerary recommendation, and logistics, the conformal set $S(A^*)$ often becomes unwieldy, containing an enor-

[1]Google Research, Mountain View, CA, USA [2]Department of Computer Science, Duke University, Durham, NC, USA. This work was done while the author was visiting Google Research. [3]Northwestern University, Evanston, IL, USA. This work was done while the author was visiting Google Research.. Correspondence to: Kamesh Munagala <kamesh@cs.duke.edu>.

*Proceedings of the 43$^{rd}$ International Conference on Machine Learning*, Seoul, South Korea. PMLR 306, 2026. Copyright 2026 by the author(s).

mous number of candidate outputs. Directly returning all elements of $S(A^*)$, such as a raw, unstructured list of thousands of potential trajectories, is impractical and leads to cognitive overload for end users. We address this by introducing *graph-based conformal compression*, a framework that summarizes the conformal set into a compact subgraph, preserving statistical coverage guarantees while minimizing structural complexity.

Unlike an explicit enumeration of paths (which can grow exponentially), a *conformal subgraph* provides a tractable and interpretable summary, even when the underlying predictive model is highly uncertain. Conversely, this framework provides a novel systems motivation for conformal prediction itself: by using past data to identify a high-probability, compressed subgraph, we safely prune the network and drastically reduce the search space for downstream routing, planning, and optimization algorithms.

Our main contribution is therefore in bridging the gap between conformal prediction and combinatorial optimization: We formally define the conformal subgraph problem and present a simple, computationally efficient algorithm that provably achieves (marginal) coverage in the distribution-free setting, while also being approximately efficient in terms of the size of the subgraph returned for that coverage bound.

In more detail, our main contributions are threefold:

**1. Algorithmic Formulation (Sections 2, 3)** We develop an algorithmic, graph-based formulation of conformal compression that treats compression as the problem of selecting a compact subgraph that contains the ground-truth output $B^*$ with high probability. We formalize the interaction between the conformal and compression guarantees in Section 2, and map the statistical goal of conformal coverage to a weighted hypergraph optimization problem in Section 3. Our framework is flexible, accommodating (i) the classic distribution-free setting (Vovk et al., 2005) with variable graph contexts (i.e., $(A_t, B_t)$ for different $t$ corresponding to different underlying graphs and source-destination pairs), (ii) its specialization to a fixed graph context (i.e., the graph and source-destination pair are the same for different $t$), (iii) model probabilities derived directly from the predictive model (Angelopoulos et al., 2022; Zhang et al., 2025) (where we assume access to a possibly mis-specified conditional distribution on $B$ given $A$), and (iv) risk-controlling prediction sets (Bates et al., 2021) to control general loss functions (e.g., controlling the expected fraction of missed route segments, or some other measure of deviation) rather than just 0-1 marginal coverage. We also show how our ideas can be applied to achieve conditional coverage (Vovk, 2012) in addition to marginal coverage.

**2. Efficiency Guarantees (Section 4).** We connect con-

formal prediction with the classical optimization problem of *densest-$k$-subgraph* selection in hypergraphs, where the goal is to select the subgraph with fewest vertices that preserves a specified fraction of the total weight of hyper-edges for a suitably defined weight function. For the resulting optimization problem, that we term the *conformal subgraph problem*, we develop provable guarantees on *efficiency*, that is, the size of the subgraph versus the conformal guarantee. We identify that conformal compression operates in a specific algorithmic regime where the subgraph retains a large fraction of hyper-edges, that is distinct from the classical, hard-to-approximate setting (Bhaskara et al., 2010; Feige et al., 2001; Manurangsi, 2017). We show that in this high-coverage regime, the problem admits an efficient bicriteria constant-factor approximation (approximating both the vertex and hyper-edge set sizes) via a simple algorithm based on linear program (LP) rounding. We show in Appendix E that this approach scales to exponentially large hyper-edge sets via sampling.

**3. Statistical Validity via Monotonicity (Section 4).** For a compression algorithm to support valid calibration in a distribution-free setting (or in the presence of a loss function (Bates et al., 2021)), it must produce a *nested* sequence of subgraphs as the target coverage varies (monotonicity). Though the optimally compressed solution for a coverage bound is not monotone in the coverage bound (see Example 3.2), we prove that our approximation algorithm satisfies monotonicity. We show this by reinterpreting our algorithm via *parametric minimum cuts* (Gallo et al., 1989; Chekuri et al., 2022), which also has the advantage of yielding an algorithm with quadratic running time. This result is one of our main contributions, and ensures that our method is a statistically valid conformal procedure for the classic distribution-free setting, both with variable and fixed graph contexts, while also enabling rigorous guarantees on both compression and validity.

We view our paper as being primarily a theoretical paper, with the main contributions being conceptual or theoretical (in establishing provable validity and efficiency guarantees for conformal compression). Nevertheless, we present a set of experiments that provide a proof of concept that the proposed method is likely to work well in practice. In Section 5 and Appendix G, we complement our theoretical results with a simulation study on the canonical applications of trip planning and noisy navigation. In the former setting, vertices are activities, hyperedges are itineraries, and a conformal subgraph is a set of activities that captures a large fraction of itineraries; while in the latter setting, vertices are graph edges, hyperedges are navigation paths, and a conformal subgraph is a subgraph that captures a large fraction of routes. In both cases, we show that our algorithm either beats natural greedy baselines on the trade-off between compression and conformal calibration, or recovers a

planted solution. We complement this with experiments on a real-world traffic dataset.

## 1.2. Related Work

**Conformal Prediction.** Conformal prediction has been applied across a wide range of areas including regression (Lei et al., 2018; Romano et al., 2019), classification (Vovk et al., 2005; Shafer & Vovk, 2007), structured models (Zhang et al., 2025), and more recently, outputs of language models (Mohri & Hashimoto, 2024; Cherian et al., 2024). Our work differs in that it seeks compact representations of conformal prediction sets themselves, rather than improving the base predictor.

There are recent extensions of conformal prediction to large or combinatorial decision problems, such as off-policy prediction (Zhang et al., 2023; Taufiq et al., 2022) and reinforcement learning (Sun et al., 2023; Strawn et al., 2023), though these works typically focus on constructing valid predictive intervals or uncertainty envelopes rather than on compressing such sets into small structural summaries.

**Efficiency in Conformal Prediction.** While conformal prediction guarantees validity by definition, minimizing the size (volume) of the prediction set, referred to as *efficiency*, is challenging, with very few recent results focusing on provable (as opposed to heuristic) efficiency of the solution (in our case, the size-optimality of the compressed graph). However, both provable validity and efficiency are a requirement, since validity is often trivial without efficiency (e.g., just output the entire graph).

Works focusing on the dual goals of conformal validity and efficiency include (Lei et al., 2013; Sadinle et al., 2019; Izbicki et al., 2020; 2022; Kiyani et al., 2024) for prediction sets and regression. The works closest to our work are (Gao et al., 2025b;a; 2026) which provide conformal guarantees with approximate size optimality in an unsupervised setting for certain families of confidence sets like balls and ellipsoids. This is related to the fixed-context setting considered in Sections 2.3 and 4.3. In contrast, in the combinatorial tasks we study, the output space is large and also precludes any posterior estimation. For instance, in routing, different calibration samples could have different source-destination pairs and different network edges. Our main contribution is developing techniques for conformal prediction in this more challenging setting that addresses both computational and statistical efficiency.

Our work is also related to recent work on structured conformal prediction (Zhang et al., 2025), which formulates conformal compression via model probabilities in the "learn–then–test" framework (Angelopoulos et al., 2022). That work constructs interpretable prediction sets by searching within a structured label family specified as a DAG. In con-

trast, we start from a calibrated conformal set, which may be large and unstructured, and develop algorithmic methods for compressing it into a small subgraph. More crucially, their approach is not monotone and hence needs model probabilities, while we obtain distribution-free calibration via showing monotonicity of the approximate compressor.

**Densest Subgraph Problems.** Our algorithms and analysis draw on a large body of work on densest subgraph discovery. The *densest ratio subgraph* problem that maximizes the average degree admits efficient polynomial-time algorithms (Goldberg, 1984; Charikar, 2000), while its fixed-size variant, the *densest-$k$-subgraph (DkS)* problem that restricts the subgraph to have at most $k$ vertices, is NP-hard and is conjectured to be inapproximable within polynomial factors (Feige et al., 2001; Bhaskara et al., 2010; Manurangsi, 2017). The *smallest-$p$-edge subgraph* (SpES) or *minimum-$p$-union* problem (Chlamtáč et al., 2018) is a complementary minimization form, with similar inapproximability results.

The strong inapproximability results for densest $k$-subgraph are proved in parameter regimes where the target subgraph has number of edges sublinear in $m$, the total number of edges. Those lower bounds therefore do not directly preclude efficient algorithms when the sought subgraph has $\Theta(m)$ edges, which is our regime of interest in conformal compression. Further, the Lagrangian of our LP formulation in Section 4 is identical to the Lagrangian for densest ratio subhypergraphs, hence reducing the problem to *parametric min-cut* on bipartite graphs (Goldberg, 1984; Gallo et al., 1989; Chekuri et al., 2022). This connection is crucial to showing monotonicity and valid calibration.

## 2. Modeling Conformal Compression

We now present the graph-based conformal compression framework, beginning with the classical distribution-free conformal calibration model. We then describe a specialization to fixed contexts, a version with model probabilities, and finally extensions to risk controlled and conditional coverage. We show that all these yield the same underlying optimization problem that we model in Section 3.

### 2.1. Distribution-free Variable-Context Setting

We adopt a split conformal approach (Lei et al., 2013; Angelopoulos & Bates, 2021) to guarantee marginal coverage without assumptions on the model distribution. Let $\mathcal{D}_{\text{cal}} = \{(A_t, B_t)\}_{t=1}^T$ denote the calibration data. For concreteness, we assume $(A_t, B_t)$ are paths in a road network, though this easily extends to other applications. Here $A_t$ is the model-predicted route and $B_t$ is the ground-truth route chosen by the user, given a context $Q_t$, comprising of a graph realization and a source-sink pair. Note that $Q_t$ can be different for different $t$. At test time, we observe

the model output $A^*$. Given a marginal coverage guarantee $\phi \in [0, 1]$, the goal is to output a set $S(A^*)$ such that $\mathbb{P}(B^* \in S(A^*)) \geq \phi$, where $B^*$ is the unknown ground truth. The probability is well-defined under an *exchangeability assumption* — the samples $\{(A_t, B_t)\}_{t=1}^T$ and $(A^*, B^*)$ are realizations of exchangeable random variables, for instance, *i.i.d.* samples from an unknown distribution.[1]

**Stage 1: Pre-Filtering.** This step mirrors classical distribution-free conformal prediction. We partition the set $\mathcal{D}_{\text{cal}}$ into two disjoint subsets, $\mathcal{D}_1$ and $\mathcal{D}_2$. Using $\mathcal{D}_1$, we calibrate the distance threshold. Let $f(A, B) \geq 0$ be a nonconformity score that measures how well a candidate route $B$ aligns with the model prediction $A$. Examples of distance functions include the number of edges on which $A$ and $B$ differ; the symmetric difference in total travel time or distance; or an embedding distance in some feature space.

Fix a parameter $\delta \ll \phi$. Now compute a threshold $d^*$ as follows: For each $(A_i, B_i) \in \mathcal{D}_1$, let

$$S_d(A_i) = \{ B : f(A_i, B) \leq d \}.$$

Now define the score

$$\gamma_i = \min\{d \geq 0 : B_i \in S_d(A_i)\}.$$

Let $d^*$ denote the $\lceil (1 - \delta)(|\mathcal{D}_1| + 1) \rceil$-th smallest value among these scores. For a new test pair $(A^*, B^*)$, this defines an initial set of plausible paths $S_{d^*}(A^*)$, and the exchangeability of $(A^*, B^*)$ with $\mathcal{D}_1$ implies (Angelopoulos & Bates, 2021; Vovk et al., 2005) that:

$$\mathbb{P}(B^* \in S_{d^*}(A^*)) \geq 1 - \delta.$$

Note now that $S_{d^*}(A^*)$ is simply a "bag of paths", an unstructured collection of trajectories that may be large[2], and the goal now is to *compress* this bag of paths into a compact subgraph while preserving a conformal guarantee.

**Stage 2: Compression.** This step is the novelty of our work. We treat this as a second filtering step parameterized by a *compression score* $\tau \in [0, 1]$. Let the set $S_{d^*}(A^*)$ consist of $M$ valid paths. We assign a utility score $w_p$ to each path $p \in S_{d^*}(A^*)$. This is typically the uniform scores $w_p = 1/M$, though the procedure below is a well-defined distribution-free conformal calibration procedure for arbitrary score assignments. We seek a subgraph of the road network that minimizes the number of edges while covering a fraction $\tau$ of the total utility score. Crucially, to preserve statistical validity, we cannot simply optimize for a fixed $\tau$. Instead, we require the following property:

---

[1]To clarify, we are not assuming the nodes or edges of the graph are exchangeable. The assumption applies to the pairs of (model prediction, ground-truth trajectory) $(A_t, B_t)$.

[2]See Appendix E for techniques to efficiently sample this set.

**Definition 2.1** (Monotonicity). An algorithm for subgraph construction is said to be *monotone* if it generates a **nested sequence** of subgraphs $K_\tau(A^*)$ such that:

$$\tau_1 < \tau_2 \implies K_{\tau_1}(A^*) \subseteq K_{\tau_2}(A^*).$$

This monotonicity allows us to treat $\tau$ as a calibration parameter. We use the second calibration split $\mathcal{D}_2$ to find the specific $\tau^*$ required to ensure the final subgraph satisfies the overall conformal guarantee $\phi$.

**Algorithm.** For each example $(A_i, B_i) \in \mathcal{D}_2$, we calculate the set $S_{d^*}(A_i)$ and run the monotone compression algorithm to obtain the sequence $\{K_\tau\}$. We compute a nested nonconformity score $\eta_i$:

$$\eta_i = \begin{cases} \min\{\tau \in [0, 1] : B_i \subseteq K_\tau(A_i)\} & \text{if } B_i \in S_{d^*}(A_i) \\ 1 & \text{if } B_i \notin S_{d^*}(A_i) \end{cases}$$

Intuitively, $\eta_i$ represents the minimum subgraph mass required to cover the ground truth. If the ground truth was lost in Stage 1 (distance $> d^*$), the score is maximal (1). We then define $\tau^*$ as the $\lceil \phi(|\mathcal{D}_2| + 1) \rceil$-th smallest value among these scores. The final prediction for a test input $A^*$ is the subgraph $K_{\tau^*}(A^*)$.

The lemma below shows the above procedure is a distribution-free conformal calibration procedure.

**Lemma 2.2** (Conformal Validity, proved in Appendix A). *The subgraph $K_{\tau^*}(A^*)$ satisfies the marginal guarantee*

$$\mathbb{P}(B^* \subseteq K_{\tau^*}(A^*)) \geq \phi - \delta.$$

The guarantee in Lemma 2.2 is strong: It serves as a distribution-free conformal wrapper around an (approximately) optimal compression of the set $S_{d^*}(A^*)$ for parameter $\tau$. It holds even when the weights $w_p$ are arbitrary and unrelated to the posterior probability $\mathbb{P}(B^* = p | A^*, B^* \in S_{d^*}(A^*))$, which in the variable context setting, may be statistically impossible to estimate (for instance, when $(A_t, B_t)_t$ for different $t$ correspond to routes for different source-sink pairs on graphs with different realized edges). On the flip side, in such a general setting, it is not possible to formally connect the conformal guarantee $\phi$ to the compression parameter $\tau$ and hence the size-optimality of $|K_\tau|$, unless we make strong assumptions. We now present two settings where this connection is possible.

### 2.2. Distribution-Based Setting

We first consider the setting where model probabilities are available, concretely the "learn then test" framework (Angelopoulos et al., 2022). Modern predictive models often output full (possibly mis-specified) conditional distributions $g(B \mid Q_t)$ rather than a single $A_t$, where $Q_t$ is the context

at step $t$. In this setting, calibration and compression can be performed directly on these probabilities, avoiding Stage 1.

For each context $Q$, let $G_\tau(Q)$ be a subgraph constructed to capture a target probability mass $\tau$, i.e., $\sum_{b \in G_\tau(Q)} g(b \mid Q) \geq \tau$. The work of (Angelopoulos et al., 2022) shows that a grid-search over $\tau$ can be used to achieve a desired conformal calibration guarantee. In this context, the compression problem is to efficiently compute $G_\tau(Q)$ as the smallest subgraph retaining probability mass $\tau$. This would constitute the ideal compression had $g$ been perfectly calibrated, since then, we would have $\phi = \tau$. Note that while the generalized learn-then-test framework can handle non-monotonicity, our specific formulation above utilizes the monotonicity property (Definition 2.1) to enable efficient quantile-based calibration without grid search.

### 2.3. Distribution-free Fixed-Context Setting

We show how our graph-based conformal compression framework yields (approximate) size-optimality guarantees along with statistical validity, in a setting that is analogous to the "unsupervised" framework of (Gao et al., 2025b). Here, we do not observe an external model output $A_t$ for varying contexts $Q_t$. Instead, we observe a history of path realizations $Y_1, \ldots, Y_T$ drawn i.i.d. from a fixed, unknown distribution $\mathcal{P}$. Such a distribution could arise if the context $Q_t$ (e.g., the graph and $s - t$ pair) remains fixed over time, which captures settings like the experiments in Section 5 and Appendix G, where we consider a fixed graph $G$ and a single, invariant source-target pair $(s, t)$. Our results apply more broadly, generalizing to any setting where we observe outcome samples $\{Y_t\}_{t=1}^T$ i.i.d. from an unknown distribution, whether or not the underlying context is fixed.

Our goal is to use this history to construct a compressed prediction set $S_{T+1}$ for the next realization $Y_{T+1}$. Rigorous guarantees on the size of the conformal prediction set are challenging to obtain in the distribution-free setting. In Section 4.3, we show how our conformal compression algorithm yields polynomial time algorithms that achieve an *approximate optimality guarantee on size* along with a conformal coverage guarantee $\phi$.

### 2.4. Other Conformal Prediction Models

Our monotonicity property (Definition 2.1) is equivalent to the nested-sets assumption required by the risk controlling prediction sets (RCPS) framework (Bates et al., 2021). This provides a statistical framework to control general loss functions (e.g., controlling the expected fraction of missed route segments, or some other measure of deviation) rather than just 0-1 marginal coverage. This is desirable since the latter can be coarse, since it does not distinguish between cases where $B^*$ is disjoint from the conformal set or almost en-

tirely overlaps with it, assigning zero score to both scenarios. The RCPS framework, via its loss function, will distinguish between these, assigning smaller loss to the second scenario. It however requires a nested sequence of sets as input. By feeding our nested conformal subgraphs $\{K_\tau\}$ into RCPS, we immediately extend our work to control arbitrary user-defined risk metrics over graphs.

Finally, our work considers marginal as opposed to conditional coverage (Vovk, 2012). For conditional coverage, as established in (Vovk, 2012; Lei et al., 2013), exact finite-sample coverage is generally impossible to achieve without making strong distributional assumptions. However, our framework is orthogonal to, and compatible with conditional coverage. For instance, if one partitions the calibration data into difficulty categories to achieve approximate conditional coverage (as proposed in (Vovk, 2012)), our algorithm can simply be applied independently within each category. This would yield compressed subgraphs that satisfy category-wise conditional coverage.

## 3. The Conformal Subgraph Problem

Both the distribution-free (Sections 2.1 and 2.3) and distribution-based (Section 2.2) formulations lead to the same underlying optimization problem: given a weighted family of candidate routes with total weight $W$, find a subgraph $G' \subseteq V$ that retains at least a $\tau = 1 - \epsilon$ fraction of $W$ while minimizing the number of edges (or another cost). The only difference lies in how the distribution over routes is obtained, via filtering on distance and assigning a utility, as in the distribution-free case, or directly from the model probabilities $g(\cdot \mid A)$ in the distribution-based case.

The subgraph we output aims to approximate the smallest subgraph achieving the desired conformal guarantee in the perfectly calibrated case. We additionally require the algorithm to satisfy monotonicity (Definition 2.1) to allow for valid calibration in the distribution-free setting.

**Problem Statement.** The conformal compression framework can therefore be expressed in combinatorial terms as follows. Each distinct candidate route $B$ in the historical data corresponds to a *hyperedge* whose weight encodes its score or model probability. The vertex set $V$ represents all atomic road segments (road edges or intersections) that may appear in any route, and each hyperedge $e \subseteq V$ corresponds to the subset of segments traversed by a particular route $B$. Let $\mathcal{E}$ denote the set of hyperedges. The nonnegative weight $w_e$ of hyperedge $e$ equals its score or its model probability, and let the total weight be $W = \sum_{e \in \mathcal{E}} w_e$.

Selecting a conformal subgraph that captures at least a fraction $\tau$ of this mass is therefore equivalent to selecting a subset of vertices $K \subseteq V$ that covers hyper-edge weight

$e(K) \geq \tau \cdot W$, where $e(S) := \sum_{\{e \in \mathcal{E} : e \subseteq S\}} w_e$. The parameter $\epsilon = 1 - \tau$ measures the fraction of probability mass allowed to remain uncovered, and the size of the chosen hypergraph vertex set reflects the number of edges in the compressed subgraph.

This abstraction leads to the following general formulation of the *conformal subgraph problem*. Let $H = (V, \mathcal{E})$ be a (possibly nonuniform) hypergraph with $|V| = n$ vertices and $|\mathcal{E}| = m$ hyperedges. Each hyperedge $e$ has a nonnegative weight $w_e$, and $W = \sum_{e \in \mathcal{E}} w_e$. Assume there exists an optimal vertex set $O \subseteq V$ satisfying: $|O| = r, e(O) \geq W - q$, and $q = \epsilon W$, so that $O$ covers all but an $\epsilon$ fraction of the total weight.

We show in Appendix B that the above problem is NP-HARD, even in the regime where $\epsilon$ is a constant. The goal is to approximate $O$ as closely as possible while allowing bicriteria trade-offs in coverage and size.

**Definition 3.1** (Bicriteria approximation). Given parameters $(\alpha, \beta)$, a vertex set $K \subseteq V$ is an $(\alpha, \beta)$-*bicriteria approximation* if $W - e(K) \leq \alpha \cdot \epsilon W$ and $|K| \leq \beta |O| = \beta \cdot r$.

**Monotonicity.** While Definition 3.1 formulates the problem for a fixed target mass $\tau = 1 - \epsilon$, the distribution-free conformal framework (Section 2) requires a nested sequence of subgraphs (see Definition 2.1). Therefore, we need our algorithm to additionally produce a family of solutions $\{K_\tau\}$ such that $K_{\tau_1} \subseteq K_{\tau_2}$ whenever $\tau_1 < \tau_2$. As shown in the example below, such a statement need not be true for the optimal solution, and one of our contributions is to show an approximation algorithm that is monotone.

**Example 3.2.** *Consider a setting with* 3 *disjoint paths* $P_1, P_2, P_3$ *between* $s$ *and* $t$. *Paths* $P_1$ *and* $P_2$ *have two edges each, while path* $P_3$ *has three edges. Suppose* 30% *of the traffic uses paths* $P_1$, 30% *uses path* $P_2$, *and* 40% *uses path* $P_3$. *If the desired coverage is* 0.6, *the optimal solution chooses paths* $P_1$ *and* $P_2$, *while if it is* 0.7, *the chosen paths are* $P_1$ *and* $P_3$, *so that the optimal solution is not monotone*.

# 4. Algorithm for Conformal Subgraphs

The previous section reformulated conformal compression as a combinatorial optimization problem on a weighted hypergraph $H = (V, \mathcal{E})$. Given an optimal (but unknown) vertex set $O$ that covers all but an $\epsilon$ fraction of the total edge weight our goal is to design an efficient algorithm that finds a small vertex subset whose induced subgraph captures nearly the same total weight.

While the general Densest $k$-Subgraph problem is hard to approximate, we show that the conformal compression regime—where the target subgraph must contain the vast majority of the total weight—allows for efficient constant-factor approximations. We further show that the resulting

algorithm satisfies monotonicity (Definition 2.1).

## 4.1. LP Rounding Algorithm

We first formulate the continuous relaxation of the problem. Let $x_v \in [0, 1]$ indicate the inclusion of vertex $v$, and $z_e \in [0, 1]$ indicate the inclusion of hyperedge $e$. Since a hyperedge is induced only if all its vertices are chosen, we enforce $z_e \leq x_v$ for all $v \in e$.

$$
\begin{aligned}
\text{minimize} \quad & \sum_{v \in V} x_v \\
\text{subject to} \quad & z_e \leq x_v, \quad \forall e \in \mathcal{E}, \forall v \in e \\
& \sum_{e \in \mathcal{E}} w_e z_e \geq (1 - \epsilon) W \\
& 0 \leq x_v, z_e \leq 1.
\end{aligned}
\tag{1}
$$

The algorithm proceeds as follows:

1. Construct and solve the linear program (1) to obtain an optimal fractional solution $(x^*, z^*)$.

2. Fix a parameter $\kappa \in \left(0, \frac{1}{\epsilon} - 1\right)$ determining the trade-off between size and coverage.

3. Return the set $K := \left\{v \in V : x_v^* \geq \rho := \frac{\kappa}{1+\kappa}\right\}$.

**Approximation analysis.** We now analyze the approximation guarantees of the LP rounding scheme. Let $r = |O|$ be the size of the optimal solution. Note that the objective value of the LP satisfies $\sum_{v \in V} x_v^* \leq r$.

**Theorem 4.1** (Proved in Appendix C). *For any* $\kappa > 0$, *the algorithm above returns a set* $K$ *that is a* $\left(1 + \kappa, \ 1 + \frac{1}{\kappa}\right)$ *bicriteria approximation: that is,*

$$
W - e(K) \leq (1 + \kappa) \epsilon W \qquad and \qquad |K| \leq \left(1 + \frac{1}{\kappa}\right) r.
$$

In the distribution-free setting (Section 2.1), this yields an approximately optimal compression of $S_{d^*}(A^*)$ while relaxing $(1 - \tau)$ by a constant factor. We interpret the guarantee for the fixed context setting from Section 2.3 in Section 4.3 below. Finally, for the distribution-based setting in Section 2.2, Theorem 4.1 directly yields an approximate optimality–coverage trade-off when $w_e$ capture the posterior probabilities of the ground truth, and this procedure can be conformalized via the learn-then-test method.

Note that though the inequality $W_{\text{lost}} := W - e(K) \leq (1 + \kappa)\epsilon W$ holds unconditionally for any graph, this bound is only meaningful in the conformal compression regime where the target subgraph retains a large fraction of edges (i.e., $\epsilon$ is a small constant). In the classical Densest $k$-Subgraph (DkS) regime, the target subgraph contains a

sublinear fraction of the edges, meaning $\epsilon \rightarrow 1$. If we plug $\epsilon \approx 1$ into our guarantee, the bound becomes $W_{\text{lost}} \leq (1+\kappa)W$, which is vacuous (it allows the algorithm to lose all edges). Therefore, while Theorem 4.1 requires no assumptions, the utility of the bicriteria approximation is mainly in the high-coverage (linear) regime, which (quite surprisingly) bypasses classical DkS hardness barriers.

### 4.2. Parametric Min-cuts and Monotonicity

We now show that our algorithm yields valid conformal guarantees in the distribution-free setting. Specifically, we prove that the solutions generated by the LP rounding form a nested family of subgraphs as the target coverage $\tau$ increases, hence satisfying Definition 2.1.

We establish this result by analyzing the Lagrangian relaxation of the LP. This Lagrangian is identical to the Lagrangian for the densest ratio sub-hypergraph, which is shown in (Chekuri et al., 2022) to map to a Parametric Minimum Cut problem (Gallo et al., 1989). The monotonicity of the Lagrangian now follows from (Gallo et al., 1989). Our subsequent contribution is to show that this monotonicity is preserved when we move from the Lagrangian to the LP optimum, and from the LP optimum to the rounding. Indeed, we show that solving the parametric minimum cut problem suffices to extract all the compressed subgraphs for various settings of the compression parameter, and these solutions are identical to those found by the LP rounding algorithm!

**Lagrangian.** In more detail, consider the Lagrangian of the LP coverage constraint with a multiplier $\lambda \geq 0$. The objective becomes:

$$\mathcal{L}(\lambda) = \min_{x,z \in [0,1]} \left( \sum_{v \in V} x_v - \lambda \sum_{e \in \mathcal{E}} w_e z_e \right)$$
$$\text{s.t.} \quad z_e \leq x_v \quad \forall e \in \mathcal{E}, \forall v \in e.$$

The above formulation is similar to the LP formulation for densest ratio subgraph in (Charikar, 2000), and their proof implies the above formulation has an integer optimum (see Lemma D.1 for a proof). We can therefore interpret the optimization variables as defining a vertex set $K = \{v \in V \mid x_v = 1\}$. Due to the constraints $z_e \leq x_v$, a hyperedge is induced ($z_e = 1$) if and only if all its vertices are in $K$. Thus, for a fixed $\lambda$, the Lagrangian minimization problem is equivalent to finding a subgraph $K \subseteq V$ that minimizes:

$$\Phi(K, \lambda) = |K| - \lambda \cdot W_{\text{induced}}(K). \tag{2}$$

**Parametric Min-cuts.** It is shown in (Chekuri et al., 2022) in the context of densest ratio sub-hypergraphs that $\mathcal{L}(\lambda)$ is precisely a standard Minimum $s - t$ Cut problem on a flow network. For this, we construct a flow network $D_\lambda$

with nodes $\{s, t\} \cup \{u_e \mid e \in \mathcal{E}\} \cup \{n_v \mid v \in V\}$ and the following edges: (1) For each hyperedge $e$, an edge $(s, u_e)$ with capacity $C_{s,u_e} = \lambda w_e$; (2) for each vertex $v$, an edge $(n_v, t)$ with capacity $C_{n_v,t} = 1$; and (3) for each inclusion $v \in e$, an edge $(u_e, n_v)$ with capacity $C_{u_e,n_v} = \infty$.

Let $K$ denote the vertices $n_v$ on the source side of the cut. Then the cut capacity is precisely $|K| + \lambda \cdot \left( \sum_{e \in \mathcal{E}} w_e - W_{\text{induced}}(K) \right)$, which shows $\Phi(K, \lambda)$ is minimized by the minimum cut.

In this graph, the set of nodes on the source side is exactly the union of the selected vertices and the induced hyperedges: $X(\lambda) = \{n_v \mid v \in K\} \cup \{u_e \mid e \text{ induced by } K\}$. The parameter $\lambda$ appears only in the capacities of the edges incident to the source $s$, and these capacities $C_{s,u_e}(\lambda) = \lambda w_e$ are non-decreasing functions of $\lambda$. This is a standard instance of the *Parametric Minimum Cut* problem, whose output is the subgraph $X(\lambda)$.

A result of (Gallo et al., 1989) for parametric min-cuts now directly implies the following lemma:

**Lemma 4.2** (Monotonicity of Lagrangian)**.** *Let $\gamma$ denote the maximum size of any hyper-edge, and let $m = |\mathcal{E}|$ and $n = |V|$. Then in $\tilde{O}(\gamma(m + n)^2)$ time, we can compute a sequence of vertex subsets $\emptyset = S_0 \subset S_1 \subset S_2 \subset \cdots \subset S_k \subseteq V$ where $k \leq n$, such that for any $\lambda \geq 0$, the solution $X(\lambda)$ is one of these subsets. Further, if $\lambda_1 < \lambda_2$, the optimal subgraphs satisfy $X(\lambda_1) \subseteq X(\lambda_2)$.*

**Algorithm.** Fix a slack parameter $\kappa$ as in the LP rounding algorithm. Given the coverage parameter $\tau := 1 - \epsilon$, the final algorithm is simple: Among the subgraphs $S_0, S_1, \ldots, S_k$ constructed above, set

$$\hat{K}_\tau = \operatorname*{argmin}_{S \in \{S_1, \ldots, S_k\}} \{|S| : W - W_{\text{induced}}(S) \leq (1 + \kappa)\epsilon W\}.$$

Note that by Lemma 4.2, we have $\hat{K}_{\tau_1} \subseteq \hat{K}_{\tau_2}$ for all $\tau_1 \leq \tau_2$, so that this construction is monotone. Further, the set of all subgraphs can be constructed in $\tilde{O}(\gamma(m + n)^2)$ time, where $\gamma$ is the largest size of a hyperedge.

Our main result is the following surprising statement, showing this algorithm is identical to the LP rounding algorithm:

**Theorem 4.3** (Proved in Appendix D)**.** *Fix a slack parameter $\kappa$. For any target coverage $\tau = 1 - \epsilon$, let $K_\tau$ be the discrete subgraph obtained by the LP rounding algorithm, and $\hat{K}_\tau$ be the subgraph obtained by the parametric min-cut algorithm. Then, $K_\tau = \hat{K}_\tau$, so that both algorithms yield the bicriteria approximation guarantee in Theorem 4.1 and produce subgraphs that are monotone in $\tau$.*

**Number of Hyperedges.** In our algorithms, we assumed the number of hyperedges $|\mathcal{E}|$ is polynomially bounded in the number of vertices $|V|$. This is a natural assumption in

many applications and it simplifies both presentation and implementation of the algorithm. In Appendix E, we show that even when the full hyperedge set output by the Pre-Filtering Stage 1 (the set $S_{d^*}(A^*)$) is exponentially large, we can replace this set by a polynomial-size random sample via the uniform convergence theorem (Vapnik & Chervonenkis, 1971), approximately preserving our guarantees. For applications like navigation, classification, and trip planning, we develop efficient sampling oracles for $S_{d^*}(A^*)$ under canonical distance functions $f(A, B)$. Uniform convergence is also relevant to our approximate optimality guarantees in the fixed context framework below.

### 4.3. Optimal Compression in Fixed Context

We now show how Theorem 4.1 can also be used to provide (approximate) size-optimality guarantees in the distribution-free conformal prediction framework in the fixed-context setting described in Section 2.3; note that this also captures the setting of our experiments on Navigation in Section 5. In this setting, the hypergraph $G(V, \mathcal{E})$ is fixed and we observe *i.i.d.* hyperedge samples $Y_1, Y_2, \ldots, Y_T$ from some unknown distribution. For given $\phi \in (0, 1)$, the goal is to output $K^*$, the subgraph satisfying the conformal guarantee $\mathbb{P}(Y_{T+1} \in K^*) \geq \phi$, while minimizing $|K^*|$.

We adapt the split conformal framework in (Gao et al., 2025b) to derive an oracle inequality for compression. Our algorithm will ensure the coverage guarantee (Validity) under just exchangeability of the samples $Y_1, \ldots, Y_{T+1}$. If in addition, the samples are i.i.d. and the number of samples is large enough, then we also get an (approximate) size optimality guarantee (Compression) by combining Theorem 4.1 with uniform convergence (Lemma E.1). Our algorithm requires a monotone compression algorithm that for any coverage parameter $\phi$, finds a subgraph whose coverage is close to that parameter. We adapt the procedure in Section 4.2 for this as follows:

1. Use half of the samples, $\{Y_j\}_{j=1}^{T/2}$, to find the nested sequence $S_0 \subset S_1 \subset \cdots \subset S_k$ as in Lemma 4.2.

2. Find the subgraph $S_i$ whose coverage on the other half of the samples, $\{Y_j\}_{j=T/2+1}^{T}$, is at least $\phi$. If $S_k$ has larger mis-coverage, add vertices to it in some fixed order till the coverage condition is satisfied.

3. Iteratively delete the vertices in $S_i \setminus S_{i-1}$ in some fixed order till the coverage becomes close to $\phi$.

**Theorem 4.4** (Proved in Appendix F). *In the fixed context setting, the above procedure yields a subgraph $\hat{K}$ satisfying:*

1. **Validity:** *If $Y_1, Y_2, \ldots, Y_{T+1}$ are exchangeable, then* $\mathbb{P}(Y_{T+1} \in \hat{K}) \geq \phi$.

2. **Approximate Compression (restates Theorem 4.1):** *Let $Y_1, Y_2, \ldots, Y_{T+1} \sim \mathcal{D}$ be i.i.d. random variables. For constants $\gamma, \kappa > 0$, let*

$$K^* = argmin_K |K| \ s.t. \ \mathbb{P}(Y_{T+1} \in K) \geq \hat{\phi} + \gamma,$$

*where $1 - \hat{\phi} = \frac{1-\phi}{1+\kappa}$. If $T = \Omega\left(\frac{|V|}{\gamma^2}\right)$, then $|\hat{K}| \leq \left(1 + \frac{1}{\kappa} + o(1)\right)|K^*|$.*

Note that the algorithm does not depend on $\kappa$, so that the compression guarantee holds simultaneously for all $\kappa > 0$. Further note that for a given $\phi$, the procedure in (Gao et al., 2025b) uses an arbitrary greedy method to construct a monotone collection of subgraphs. Our construction via Theorem 4.3 is more robust when the sample size $T$ is much smaller than the uniform convergence threshold, so that a theoretical bound on compression is not possible. This aspect is particularly relevant to our experiments in Appendix G.1 on real-world data that can be sparse.

## 5. Empirical Results on Navigation

In this section, we evaluate the efficacy of our graph-based conformal compression framework. Our focus is on the combinatorial optimization challenge mirroring the fixed context setting (Section 2.3): compressing a large, complex set of candidate outputs (hyperedges) into a compact subgraph. We consider two settings: a *Navigation* setting based on road networks to study the trade-off between conformal calibration and compression, and a *Trip Planning* simulation designed to test the algorithm's ability to recover planted structures. We present the synthetic navigation setting below, and prent the real-world Navigation experiment and the trip planning experiment in Appendix G.

**Navigation Setting.** In this setting, hyperedges correspond to routing paths in the graph, and hypergraph vertices correspond to edges in the graph. We compare our LP-based algorithm against *greedy*, which select graph edges strictly order of traffic frequency (number of paths).

We consider both the ability of the algorithm to compress a set of routes by comparing to a greedy baseline, as well as the conformal coverage $\phi$ that ensues. We study the latter in the split conformal framework of Section 4.3: We generate hyper-edge samples $\{Y_j\}_{j=1}^{T/2}$ (denoted "training") and $\{Y_j\}_{j=T/2+1}^{T}$ (denoted "test") from the same underlying model. We use the procedure in Section 4.3 to find a nested sequence of subgraphs on the training samples, and choose a subgraph as described there to obtain coverage $\phi$ on the test samples. Note that by Theorem 4.3, the nested subgraph sequence is fixed and oblivious to the parameter $\kappa$ (which is only needed to state Theorem 4.4). Further, the number of sampled routes in our experiments is small enough that the LP method runs in under a minute for our instances.

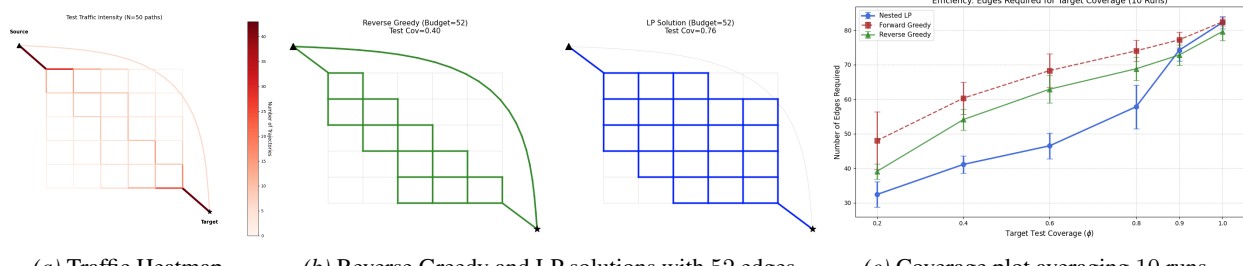

*(a)* Traffic Heatmap.      *(b)* Reverse Greedy and LP solutions with 52 edges.      *(c)* Coverage plot averaging 10 runs.

*Figure 1.* Analysis of the synthetic urban routing scenario.

**Greedy Baselines.** We consider two greedy baselines. In FORWARD GREEDY, we sort edges in decreasing order of number of training paths they cover, and choose them in this order till a desired coverage $\phi$ is achieved on the test samples.The REVERSE GREEDY algorithm is a generalization of the greedy vertex deletion algorithm in (Charikar, 2000) to hypergraphs. Specialized to routing, it deletes edges in reverse order of traffic intensity (number of training paths); however, when it deletes an edge, it deletes all training paths passing through it, and recomputes traffic intensity on the left-over edges. We repeat till the coverage on the test samples falls below $\phi$, stopping just before that. Such greedy algorithms are clearly monotone and will therefore yield conformal guarantees analogous to the LP method. However, as we show in Appendix G.1, they will not provide a compression guarantee analogous to Theorem 4.1.

We will compare the number of edges used by the LP and greedy solutions for different values of $\phi$. This allows us to study the calibration-compression trade-off without the need to simulate a predictive model.

**Synthetic Routing Experiment.** This simulation tests a realistic routing scenario where traffic between two points either takes a fast, but long bypass, or takes a city grid between the two points. We simulate a $6 \times 6$ grid. Traffic flows from a single source at the top-left to a single target at the bottom right. We use this grid to sample $85\%$ of the paths using shortest-paths with random edge weights, where the weight on an edge is `Uniform`$[0.1, 2]$. There is a bypass route from the source to the target with 20 edges that is taken by $15\%$ of the traffic. We generate a calibration set (train) and held out (test) set of 50 routes. The traffic heatmap is shown in Figure 1(a). The paths spread out in the middle of the grid, leading to lower traffic intensity there.

Such a setting is not only the simplest instantiation of routing, but is also quite realistic for routing in urban areas such as Manhattan, where streets can get congested unpredictably, but there are a plethora of alternate routes.

We implement the LP and greedy calibration procedures as described above. Figure 1(c) shows the number of edges

used by the LP and greedy solutions, where the $x$-axis shows the test coverage $\phi$. Note that the LP solution is significantly more compressed than either greedy algorithm for all $\phi \leq 0.8$. In Figure 1(b), we show the edge selections when the LP solution chooses 52 edges ($\phi = 0.75$) and the reverse greedy algorithm is made to choose the same number. The reverse greedy algorithm prioritizes the highway edges, while the LP correctly identifies that the urban center, while consisting of individually lower-frequency edges, is needed for coverage, and omits the highway entirely.

## 6. Conclusion

Our framework opens several avenues for future research. First, many practical domains exhibit specific hypergraph structures. For instance, road networks are nearly planar, and hierarchical classification labels form trees. It is an open question whether the conformal subgraph problem admits tighter approximations (e.g., PTAS) or exact solutions when restricted to these special graph classes. Similarly, real-world predictive distributions often exhibit planted structure, like the trip planning scenario in Appendix G.2. Can we prove tighter bounds or exact recovery guarantees for our algorithms under these generative assumptions?

Stepping back, we are treating the predictive model as fixed. However, conformal validity holds for *any* scoring function. This allows for an end-to-end approach where the model $f_\theta$ is trained to explicitly optimize for compressibility, One could, for instance, augment the training objective with a differentiable "graph compression" regularizer. However, such regularizers will introduce significant non-convexity, addressing which forms an interesting research direction.

**Use of Generative AI.** We used Gemini 3 Pro for: (1) identifying and summarizing prior work; (2) paraphrasing and polishing the text; and (3) the development of the synthetic and real-world navigation experiments, which included understanding the structure of the Porto dataset, and implementing the Python code. All AI-generated content, particularly citations and code logic, was verified by the authors, who take full responsibility for its correctness.

## Impact Statement

This paper's goal is to advance the field of Machine Learning, specifically in the area of uncertainty quantification in structured prediction. There are many potential societal consequences of our work, none which we feel must be specifically highlighted here.

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

## A. Proof of Lemma 2.2

We analyze the probability space defined by the randomness of the calibration splits $\mathcal{D}_1, \mathcal{D}_2$ and the test point $(A^*, B^*)$.

First, condition on the realization of $\mathcal{D}_1$. Since $\mathcal{D}_1$ and $\mathcal{D}_2$ are disjoint, the threshold $d^*$ (computed solely on $\mathcal{D}_1$) is treated as a fixed constant with respect to $\mathcal{D}_2$. Consequently, the mapping from any input-output pair $(A, B)$ to its nested nonconformity score $\eta$ depends only on deterministic functions (the fixed model, the metric $f$, the fixed threshold $d^*$, and the deterministic compression algorithm).

Under the standard exchangeability assumption, the scores $\{\eta_i\}_{i \in \mathcal{D}_2}$ computed on the second calibration split and the score of the test point $\eta^*$ are exchangeable random variables. By the standard property of conformal prediction quantiles (Vovk et al., 2005), choosing $\tau^*$ as the $\lceil \phi(|\mathcal{D}_2| + 1) \rceil$-th smallest score guarantees:

$$\mathbb{P}(\eta^* \leq \tau^* \mid \mathcal{D}_1) \geq \phi.$$

Since this holds for any realization of $\mathcal{D}_1$, we have

$$\mathbb{P}(\eta^* \leq \tau^*) \geq \phi. \tag{3}$$

Next, we analyze the relationship between the score condition $\eta^* \leq \tau^*$ and the coverage event $B^* \subseteq K_{\tau^*}(A^*)$. Recall the definition of the score $\eta^*$:

- If $B^* \in S_{d^*}(A^*)$, then $\eta^*$ is the smallest $\tau$ such that $K_\tau$ covers $B^*$. Thus, $\eta^* \leq \tau^*$ implies $B^* \subseteq K_{\tau^*}(A^*)$.

- If $B^* \notin S_{d^*}(A^*)$ (Stage 1 failure), then $\eta^* = 1$. In this case, the condition $\eta^* \leq \tau^*$ can only be satisfied if $\tau^* = 1$. Even if $\tau^* = 1$, the subgraph $K_1$ cannot contain $B^*$. To see this, note that if the ground-truth $B^*$ was lost in Stage 1 $(B^* \notin S_{d^*})$, it means $B^*$ utilizes at least one road segment that is not in $S_{d^*}(A^*)$. Therefore, no subset of $S_{d^*}(A^*)$ (not even the maximal subset $K_1$) can fully cover $B^*$.

Therefore, the event $\{\eta^* \leq \tau^*\}$ is equivalent to the union of two disjoint events:

$$\{\eta^* \leq \tau^*\} \iff \{B^* \subseteq K_{\tau^*}(A^*)\} \cup \{B^* \notin S_{d^*}(A^*) \wedge \tau^* = 1\}.$$

Rearranging this probability:

$$\mathbb{P}(B^* \subseteq K_{\tau^*}(A^*)) = \mathbb{P}(\eta^* \leq \tau^*) - \mathbb{P}(B^* \notin S_{d^*}(A^*) \wedge \tau^* = 1).$$

By the exchangeability of $(A^*, B^*)$ with $\mathcal{D}_1$, the threshold $d^*$ computed in the pre-filtering stage ensures:

$$\mathbb{P}(B^* \notin S_{d^*}(A^*)) \leq \delta.$$

This implies the joint event is also bounded:

$$\mathbb{P}(B^* \notin S_{d^*}(A^*) \wedge \tau^* = 1) \leq \mathbb{P}(B^* \notin S_{d^*}(A^*)) \leq \delta.$$

Combining this with Eq. (3):

$$\mathbb{P}(B^* \subseteq K_{\tau^*}(A^*)) \geq \phi - \delta.$$

This completes the proof. $\qquad\square$

## B. NP-Hardness of the Conformal Subgraph Problem

Our main result in this section is to show that the conformal subgraph problem is NP-HARD even in the regime where $\epsilon$ is a constant.

**Theorem B.1.** *Fix any constant $0 < \epsilon < 1$. The decision problem*

> *Given a graph $G' = (V', E')$, $k \in \mathbb{N}$, is there $K \subseteq V'$ with $|K| \leq k$ and $e_{G'}(K) \geq (1 - \epsilon)W$?*

*(where $W = |E'|$) is* NP-HARD.

*Proof.* Let $(G = (V, E), r)$ be an arbitrary CLIQUE instance with $n = |V|$, $m = |E|$, and maximum degree $d = \max_{v \in V} \deg_G(v)$. A YES instance has a clique of size $r$, while a NO instance has maximum clique size less than $r$.

We construct $G'$ as follows. Form $G_c$ as the disjoint union of $c$ identical copies of $G$. The number of edges in $G_c$ is $m_c := cm$, while the per-vertex degree remains at most $d$. Add a disjoint gadget clique $C$ on $s$ new vertices. Finally, add a set $P$ of disjoint "padding" edges (isolated edges that do not touch $G_c$ or $C$).

Let $W$ be the total number of edges in $G'$. We define the target vertex budget $k' := r + s$. We define the edge count of a solution that includes the gadget and an $r$-clique as:

$$L_{YES} := \binom{s}{2} + \binom{r}{2}.$$

If the instance is a NO instance, any set of $r$ vertices from $G_c$ induces at most $\binom{r}{2} - 1$ edges. Thus, a solution with the gadget and $r$ vertices from $G_c$ would have at most $L_{YES} - 1$ edges.

We choose the parameters $c$, $s$, and the padding size $|P|$ so that the following three properties hold:

(P1) The threshold separates the YES and NO cases. We require $W$ to satisfy:

$$L_{YES} - 1 \;<\; (1 - \epsilon)W \;\leq\; L_{YES}. \tag{4}$$

This ensures that obtaining $L_{YES}$ edges is sufficient, but obtaining $L_{YES} - 1$ is not.

(P2) Omitting even a single gadget vertex loses so many gadget edges that the lost edges cannot be compensated by adding available extra vertices from $G_c$ (whose individual degrees are $\leq d$). Concretely, it suffices to enforce

$$s - 1 \;>\; 2d. \tag{5}$$

(P3) The graph construction is valid. We need the required total weight $W$ to be at least the number of edges in the main components $(m_c + \binom{s}{2})$, so that the number of padding edges $|P| = W - (m_c + \binom{s}{2})$ is non-negative.

**Parameter Selection.** We assume $m$ is large. We perform the choices in the following order:

1. For sufficiently large $c$, choose $s$ as the smallest integer such that $\epsilon \cdot \binom{s}{2} \geq (1 - \epsilon)m_c$. Since the RHS grows with $c$ while $d = O(n)$, for polynomially large $c$ the condition $s > 2d + 1$ from (5) is satisfied.

2. Rearranging (4), we need an integer $W$ in the interval $\left(\frac{L_{YES}-1}{1-\epsilon}, \frac{L_{YES}}{1-\epsilon}\right]$. The length of this interval is $\frac{1}{1-\epsilon}$. Since $0 < \epsilon < 1$, the length is strictly greater than 1. Therefore, such an integer $W$ exists.

3. Set the number of padding edges $|P| = W - (m_c + \binom{s}{2})$. We must verify that $|P| \geq 0$. From the choice of $W$, we know $W > \frac{L_{YES}-1}{1-\epsilon}$. Since $r \geq 2$, $\binom{r}{2} \geq 1$, and thus $L_{YES} - 1 = \binom{s}{2} + \binom{r}{2} - 1 \geq \binom{s}{2}$. Therefore, $W \geq \frac{1}{1-\epsilon}\binom{s}{2}$. From the choice of $s$, we have $\epsilon\binom{s}{2} \geq (1 - \epsilon)m_c$. Adding $(1 - \epsilon)\binom{s}{2}$ to both sides yields $\binom{s}{2} \geq (1 - \epsilon)(m_c + \binom{s}{2})$, or equivalently $\frac{1}{1-\epsilon}\binom{s}{2} \geq m_c + \binom{s}{2}$. Combining these inequalities gives $W \geq m_c + \binom{s}{2}$, ensuring $|P| \geq 0$.

**Correctness.** First, if $G$ contains an $r$-clique $Q$, let $K = V(C) \cup Q$. Then $|K| = s + r = k'$. The induced edges are $e(K) = \binom{s}{2} + \binom{r}{2} = L_{YES}$. By the RHS of (4), $L_{YES} \geq (1 - \epsilon)W$. Thus, this is a YES instance. Conversely, suppose there exists $K' \subseteq V(G')$ with $|K'| \leq k'$ and $e_{G'}(K') \geq (1 - \epsilon)W$. Note that padding edges are isolated and disjoint; selecting vertices incident to padding edges is strictly suboptimal compared to selecting vertices in $C$ (degree $s - 1$) or $G_c$ (degree up to $d$), so we assume $K' \subseteq V(G_c) \cup V(C)$. Let $x = |K' \cap V(C)|$ and $y = |K' \cap V(G_c)|$, so $x + y \leq r + s$.

- **Case 1:** $x = s$. All gadget vertices are chosen. Then $y \leq r$. The edges induced are $\binom{s}{2} + e_{G_c}(K' \cap V(G_c))$. Since $e_{G'}(K') \geq (1 - \epsilon)W$, by the LHS of (4) we must have $e_{G'}(K') > L_{YES} - 1$. Therefore, the $y$ vertices in $G_c$ must induce $> \binom{r}{2} - 1$ edges. This implies they induce at least $\binom{r}{2}$ edges. Thus, $G_c$ contains an $r$-clique.

- **Case 2:** $x \le s - 1$. Let $z \ge 1$ be the number of gadget vertices omitted from the final subgraph. The number of edges lost from the gadget is $\binom{s}{2} - \binom{s-z}{2}$, which is at least $\frac{z(s-1)}{2}$. If we choose $z$ extra vertices from $G_c$ (beyond the $r$ allowed), the number of edges gained there is at most $z \cdot d$ (since every vertex in $G_c$ has degree at most $d$). By (5), we have $s - 1 > 2d$, which implies $\frac{z(s-1)}{2} > z \cdot d$. Thus, the loss from the gadget strictly exceeds the gain from $G_c$. Any such subgraph with $x + y$ vertices therefore has strictly fewer than $L_{YES} = \binom{s}{2} + \binom{r}{2}$ edges. Consequently, the weight of this subgraph is strictly below $(1 - \epsilon)W$ and this case is impossible.

The reduction is poly-time as $c$ is polynomial in the input size. This proves NP-HARDNESS for any fixed constant $\epsilon$. $\square$

## C. Proof of Theorem 4.1

By definition, every vertex $v \in K$ has fractional value $x_v^* \ge \rho$. Thus,

$$|K| = \sum_{v \in K} 1 \le \sum_{v \in K} \frac{x_v^*}{\rho} \le \frac{1}{\rho} \sum_{v \in V} x_v^* \le \frac{1}{\rho} r. \tag{6}$$

Substituting $\rho = \frac{\kappa}{1+\kappa}$, we obtain the size bound on $|K|$.

Next, we bound the weight of the edges *not* covered (lost) by $K$. Let $W_{\text{lost}} = W - e(K)$ be the total weight of hyperedges not induced by $K$. A hyperedge $e$ is not induced in $K$ if and only if at least one of its vertices $v \in e$ is excluded from $K$ (i.e., $x_v^* < \rho$). From the LP constraints, for every edge $e$, we have $z_e^* \le \min_{v \in e} x_v^*$. Therefore, if edge $e$ is not induced in $K$, it must satisfy $z_e^* < \rho$. Since $1 - z_e^* \ge 0$, we have:

$$W_{\text{lost}} = \sum_{e \not\subseteq K} w_e < \sum_{e \not\subseteq K} w_e \left( \frac{1 - z_e^*}{1 - \rho} \right) \le \frac{1}{1 - \rho} \sum_{e \in \mathcal{E}} w_e (1 - z_e^*).$$

The LP constraint implies $\sum w_e (1 - z_e^*) \le \epsilon W$. Substituting this into the inequality above:

$$W - e(K) = W_{\text{lost}} \le \frac{1}{1 - \rho} (\epsilon W) = (1 + \kappa) \epsilon W.$$

This completes the proof. $\square$

## D. Proof of Theorem 4.3

Recall the LP formulation in Section 4. We wish to minimize the total vertex mass $\sum x_v$ subject to the coverage constraint $\sum w_e z_e \ge \tau \cdot W$ and the constraints $z_e \le x_v$. As discussed in Section 4.2, consider the Lagrangian of the coverage constraint with a multiplier $\lambda \ge 0$. The objective becomes:

$$\mathcal{L}(\lambda) = \min_{x,z \in [0,1]} \left( \sum_{v \in V} x_v - \lambda \sum_{e \in \mathcal{E}} w_e z_e \right) \quad \text{s.t.} \quad z_e \le x_v \quad \forall e \in \mathcal{E}, \forall v \in e. \tag{7}$$

We first have the following, which follows from an analogous result for densest ratio subgraphs:

**Lemma D.1** (Integrality of the Lagrangian (Charikar, 2000)). *For any $\lambda \ge 0$, there exists an optimal solution $(x^*, z^*)$ to $\mathcal{L}(\lambda)$ such that $x_v^* \in \{0, 1\}$ for all $v$ and $z_e^* \in \{0, 1\}$ for all $e$.*

*Proof.* Pick a threshold $\alpha \in [0, 1]$ uniformly at random, and set $x_v = 1$ if $x_v \ge \alpha$ (and 0 otherwise). Because the fractional LP forces the hyperedge variable $z_e \le \min_{v \in e} x_v$, setting the vertices this way ensures the constraints are preserved, and the Lagrangian objective is preserved in expectation. This implies the existence of an integer solution matching the fractional value. $\square$

### D.1. Structure of the LP Solution

As shown in Section 4.2, the Lagrangian is a parametric min-cut problem with an integer optimum, and by Lemma 4.2, the optimal solution $X(\lambda)$ is nested in $\lambda$. Having established that the integer solutions to the Lagrangian relaxation are nested, we now characterize the optimal fractional solution $x^*(\tau)$ to the LP in Section 4 for a specific target coverage $\tau$.

**Lemma D.2** (Structure of LP Solution). *Let $\tau$ be a target coverage. There exists a critical Lagrange multiplier $\lambda^* \geq 0$ and two vertex sets $S^-$ and $S^+$ such that:*

1. *$S^-$ and $S^+$ are both optimal integral solutions to the Lagrangian relaxation at $\lambda^*$.*

2. *$S^-$ is nested within $S^+$ (i.e., $S^- \subseteq S^+$).*

3. *The optimal fractional LP solution $x^*(\tau)$ is a convex combination:*

$$x^*(\tau) = (1 - \alpha)\mathbf{1}_{S^-} + \alpha\mathbf{1}_{S^+}$$

   *where $\alpha$ is chosen such that the weight constraint $\sum_e w_e \cdot z_e \geq \tau \cdot W$ is met.*

*Proof.* Let $L(\lambda) = \min_K(|K| - \lambda W_{\text{induced}}(K))$ be the Lagrangian dual function. By strong duality, there exists a $\lambda^*$ such that the optimal LP value is $L(\lambda^*) + \lambda^* \tau \cdot W$, where $W = \sum_{e \in \mathcal{E}} w_e$. This $\lambda^*$ corresponds to a "breakpoint" in the piecewise-linear concave function $L(\lambda)$, where the subgradient contains $\tau$.

To define $S^-$ and $S^+$, we consider infinitesimal perturbations of $\lambda^*$: Let $S^-$ be an optimal solution for $\lambda^- = \lambda^* - \delta$ for arbitrarily small $\delta > 0$, and let $S^+$ be an optimal solution for $\lambda^+ = \lambda^* + \delta$. By Lemma 4.2, we have $S^- \subseteq S^+$. By the continuity of the objective function, as $\delta \to 0$, $S^-$ and $S^+$ must also be optimal for the exact parameter $\lambda^*$. Specifically:

$$|S^-| - \lambda^* W(S^-) = |S^+| - \lambda^* W(S^+) = L(\lambda^*).$$

Since $\lambda^*$ is a breakpoint where the subgradient includes $\tau \cdot W$, we have $W(S^-) \leq \tau \leq W(S^+)$. The vector $\hat{x} = (1 - \alpha)\mathbf{1}_{S^-} + \alpha\mathbf{1}_{S^+}$ is a convex combination of two optimal integral solutions, so it is a valid solution to the Lagrangian relaxation. By setting $\alpha = \frac{\tau \cdot W - W(S^-)}{W(S^+) - W(S^-)}$, the solution $\hat{x}$ satisfies the coverage constraint $\sum w_e z_e = \tau \cdot W$ with equality. Substituting this into the primal objective:

$$\begin{aligned}
\text{Size}(\hat{x}) &= (1-\alpha)|S^-| + \alpha|S^+| = (1-\alpha)(L(\lambda^*) + \lambda^* W(S^-)) + \alpha(L(\lambda^*) + \lambda^* W(S^+)) \\
&= L(\lambda^*) + \lambda^*\big((1-\alpha)W(S^-) + \alpha W(S^+)\big) = L(\lambda^*) + \lambda^* \tau \cdot W.
\end{aligned}$$

Thus, $\hat{x}$ is the optimal primal solution. $\square$

### D.2. The Rounding Procedure

Finally, we show that the LP rounding algorithm is identical to the parametric min-cut algorithm in terms of the subgraphs generated, hence showing both algorithms satisfy Theorem 4.1, and are monotone.

**Lemma D.3** (Structure of Threshold Rounding). *Let $x^*(\tau)$ be the optimal fractional solution for target coverage $\tau$. Define the vertex set $K_\tau$ by including all vertices where the fractional value exceeds a fixed threshold $\rho \in (0, 1]$ (i.e., $K_\tau = \{v \in V \mid x_v^*(\tau) \geq \rho\}$). Then, each $K_\tau$ is the optimal solution to $\mathcal{L}(\lambda)$ for some $\lambda$. Further, the sequence of sets is nested:*

$$\tau_1 < \tau_2 \implies K_{\tau_1} \subseteq K_{\tau_2}.$$

*Proof.* From the proof of Lemma D.2, the optimal sets for the Lagrangian can be chosen to be nested such that $S^- \subseteq S^+$. Consider the behavior of the variable $x_v^*(\tau)$ as $\tau$ increases. The algorithm progresses through the nested chain of sets $S_0 \subset S_1 \subset \ldots$. Within any interval $(W(S_i), W(S_{i+1})]$, the solution is given by $x^* = (1 - \alpha)\mathbf{1}_{S_i} + \alpha\mathbf{1}_{S_{i+1}}$. The value of $x_v^*$ for a specific vertex $v$ behaves as follows:

- If $v \in S_i$, then $v \in S_{i+1}$ as well. Thus $x_v^* = 1$.

- If $v \in S_{i+1} \setminus S_i$, then $x_v^* = \alpha$. Since $\tau = (1 - \alpha)W(S_i) + \alpha W(S_{i+1})$, $\alpha$ is strictly increasing with $\tau$.

- If $v \notin S_{i+1}$, then $x_v^* = 0$.

Observe now that for any $\tau$, this implies the LP variables are fractional for vertices contained in some $S_2$ but not in $S_1 \subset S_2$, where $S_1$ and $S_2$ are both optimal solutions to $\mathcal{L}(\lambda^*)$ for some $\lambda^*$. Further, these fractional values are all the same. This means threshold rounding either chooses $S_1$ or $S_2$ as $K_\tau$, so that $K(\tau) = \mathcal{L}(\lambda)$ for some $\lambda$.

Finally, note that as $\tau$ moves to the next interval, the sets $S^-$ and $S^+$ change to $S_{i+1}$ and $S_{i+2}$, maintaining the non-decreasing property. Thus, for every vertex $v$, $x_v^*(\tau)$ is a non-decreasing function of $\tau$. This ensures that the LP solution is monotone. Consider any vertex $v \in K_{\tau_1}$. By definition, this implies $x_v^*(\tau_1) \geq \rho$. Since $\tau_2 > \tau_1$ and $x_v^*(\cdot)$ is non-decreasing, we have $x_v^*(\tau_2) \geq x_v^*(\tau_1) \geq \rho$. Consequently, $v$ satisfies the threshold condition for $\tau_2$ and is included in $K_{\tau_2}$. Since this holds for any $v$, we conclude $K_{\tau_1} \subseteq K_{\tau_2}$. This completes the proof. $\qquad\square$

**Proof of Theorem 4.3.** This directly follows from Lemma D.3. $\qquad\square$

# E. Sampling Hyperedges and Uniform Convergence

## E.1. Uniform Convergence

In applications where the hyperedge family $\mathcal{E}$ is too large to process directly, we replace $\mathcal{E}$ by an i.i.d. sample $\widehat{\mathcal{E}} = \{\widehat{e}_1, \ldots, \widehat{e}_{\hat{m}}\}$ from the distribution over hyperedges that places probability mass proportional to edge weight $w_e$ on hyperedge $e$. All steps in the conformal compression algorithm in Section 4 can then be executed on the sample. The following statement quantifies the sample size needed so that every vertex subset's induced mass is approximated uniformly.

**Lemma E.1** (Uniform convergence (Devroye & Lugosi, 2001), Chapter 4). *Let $H = (V, \mathcal{E})$ be a hypergraph with $|V| = n$. For every vertex subset $S \subseteq V$ define the induced total population mass $e(S) := \sum_{e \in \mathcal{E}: e \subseteq S} w_e$ and total mass $W := \sum_{e \in \mathcal{E}} w_e$. Let $\widehat{\mathcal{E}}$ be an i.i.d. sample of $\hat{m}$ hyperedges drawn from the distribution that chooses $e$ with probability $w_e/W$. Fix $\alpha, \delta \in (0, 1)$. There exists an absolute constant $C$ such that if*

$$\hat{m} \geq C \cdot \frac{\mathrm{VC}(\mathcal{F}) + \log(1/\delta)}{\alpha^2},$$

*where $\mathcal{F} = \{f_S : f_S(e) = \mathbf{1}\{e \subseteq S\}, S \subseteq V\}$, then with probability at least $1 - \delta$ (over the draw of $\widehat{\mathcal{E}}$) the following holds simultaneously for every $S \subseteq V$:*

$$\left| \frac{\sum_{\widehat{e} \in \widehat{\mathcal{E}}} \mathbf{1}\{\widehat{e} \subseteq S\}}{\hat{m}} - \frac{e(S)}{W} \right| \leq \alpha.$$

*In particular, since $\mathrm{VC}(\mathcal{F}) \leq n$, it suffices to take $\hat{m} = O\big((n + \log(1/\delta))/\alpha^2\big)$.*

Lemma E.1 implies that for every vertex set $S$ the *empirical* induced mass measured on $\widehat{\mathcal{E}}$ approximates the true induced mass $e(S)/W$ up to additive $\alpha$. Consequently, running the algorithm of Section 4 on the sampled hyperedges produces a vertex set $\widehat{K}$ whose empirical retained mass is (by that algorithm) at least the target threshold. By uniform convergence, the *true* retained mass $e(\widehat{K})/W$ is within an additive $\alpha$ of the empirical value, and thus the bicriteria bound proved for the algorithm on the full hyperedge set degrades only by an additive $O(\alpha)$ factor when executed on the sample. In particular, for any desired additive tolerance $\alpha > 0$ one can choose $\hat{m} = O((n + \log(1/\delta))/\alpha^2)$ so that the sample solution satisfies the same size vs. retained-mass bicriteria guarantee as in Theorem 4.1 up to an additive $\alpha$ loss, with probability $\geq 1 - \delta$. Therefore, in the perfectly calibrated case of the weights, this preserves the approximation guarantees for compression to an additive $O(\alpha)$.

In terms of calibration, sampling hyper-edges to approximate the compressor does not alter the exchangeability required for either distribution-free calibration or the "learn–then–test" conformal guarantee of Section 2. The sampled compressor remains a function solely of the model and input $Q$, independent of the calibration labels $B$, so the scores remain exchangeable, and the compressed graph remains monotone.

## E.2. Efficient Sampling Oracle for the Set $S_d(A)$

To apply the uniform convergence result (Lemma E.1), we require a computationally efficient oracle that can sample hyper-edges at random from the conformal set $S_{d^*}(A^*)$. In this section, we assume the sampling weights are uniform.

### E.2.1. SAMPLING WALKS VIA DYNAMIC PROGRAMMING

We first consider the setting of $s$–$t$ routing with uniform weights, where the underlying graph has $n$ vertices and $m$ edges. For efficient sampling, we make two assumptions on the conformal set: First, we assume the set $S_{d^*}(A^*)$ can include walks and not just paths. Second, we assume $A^*$ is a simple path directed from $s$ to $t$, and for any $B \in S_{d^*}(A^*)$, the edges in $A^* \cap B$ are directed in $B$ in the same way as in $A^*$.

Under these assumptions, we direct the edges in $A^*$ from $s$ to $t$ in $E$, while leaving the remaining edges undirected. Any valid $B \in S_{d^*}(A^*)$ is a $s$–$t$ walk on this graph. Now treat $B$ as a multi-set of edges and consider the distance measure $f(A^*, B)$ as the number of edges in the multiset that do not belong to $A^*$.

To interpret this, assign binary weights to the edges of the graph $G = (V, E)$ (with edges in $A^*$ directed as above), so that for $(u, v) \in A^*$, $w_{uv} = 0$, while for $(u, v) \notin A^*$, $w_{uv} = 1$. Under this weighting, the condition $f(A^*, B) \leq d^*$ is equivalent to the condition that the total weight of the $s$–$t$ walk $B$ is at most $d^*$.

Let $N(u, k)$ denote the number of walks from node $u$ to the target $t$ with total accumulated weight exactly $k$. The base case is $N(t, 0) = 1$ (and 0 for $k > 0$). The recurrence relation is:

$$N(u, k) = \sum_{(u,v) \in E} N(v, k - w_{uv}).$$

In the above recurrence, we assume the edges in $A^*$ are directed from $s$ to $t$ in $E$. Since edges have weights of either 0 or 1, it is now easy to check that we can compute the table iteratively in increasing order of $k$, and for each $k$, first in reverse order on the vertices in the path $A^*$, and then arbitrarily for the other vertices. This is because $N(u, k)$ for $u$ lying on $A^*$ depends on $N(v, k)$ if $(u, v) \in A^*$ and on $N(v, k - 1)$ if $(u, v) \notin A^*$, while $N(u, k)$ for $u$ outside $A^*$ depends on $N(v, k - 1)$ for edges $(u, v) \in E$. This procedure therefore computes exact counts in $O(d^* \cdot |E|)$ time.

Once the table $N(u, k)$ is populated, sampling a walk uniformly from $S_{d^*}(A^*)$ is reduced to a stateless random walk. First, we determine the total number of valid paths $Z = \sum_{j=0}^{d^*} N(s, j)$. We sample a target budget $K \in \{0, \ldots, d^*\}$ with probability $N(s, K)/Z$. We then construct the path node-by-node starting at state $(s, K)$. Given current state $(u, k)$, we consider states $(v, k')$ that contribute to $N(u, k)$ in the above recurrence, and transition to one of them with probability proportional to $N(v, k')$. We repeat this till $u = t$. This procedure generates independent, identically distributed samples from the uniform distribution over $S_{d^*}(A^*)$, satisfying the requirements for the uniform convergence bound.

### E.2.2. EXTENSION TO OTHER SETTINGS

This sampling approach generalizes to any structured output space that admits a dynamic programming decomposition. As an example, consider *hierarchical classification* (also considered in (Zhang et al., 2025)) where the label space is a tree $\mathcal{T}$ and a hyperedge corresponds to a valid subtree rooted at the origin (e.g., a pruned classification path). Given a model-output subtree $A^*$, let $S_{d^*}(A^*)$ define all subtrees rooted at the origin that have at most $d^*$ vertices not in $A^*$. As before, we assign weights to vertices such that $w_v = 1$ if $v \notin A^*$ and 0 otherwise. We define the DP state $N(u, k)$ as the number of valid subtrees rooted at $u$ that contain exactly $k$ nodes not present in $A^*$. The recurrence relation becomes a convolution of the counts from the children of $u$: if $u$ has children $v_1, \ldots, v_m$, then

$$N(u, k) = \sum_{j_1 + \cdots + j_m = k - w_u} \prod_{i=1}^{m} N(v_i, j_i).$$

Using standard DP techniques, this convolution can be computed efficiently for any $k$. Using the random walk method from Step (3) above, we can now generate an exact uniform sample of hierarchical conformal sets.

As a final example, consider the *Trip Planning* setting from Section 5 where the universe of activities is partitioned into disjoint groups $G_1, \ldots, G_R$ (activity types), and a valid hyperedge consists of exactly one activity from each group. Given a reference itinerary $A^*$, let $S_{d^*}(A^*)$ denote the set of itineraries that have at most $d^*$ activities not in $A^*$. We assign weight 0 to activities in $A^*$ and 1 to all others. We define $N(r, k)$ as the number of valid partial itineraries using groups $r$ through $R$ that accumulate exactly cost $k$. We have:

$$N(r, k) = \sum_{v \in G_r} N(r + 1, k - w_v).$$

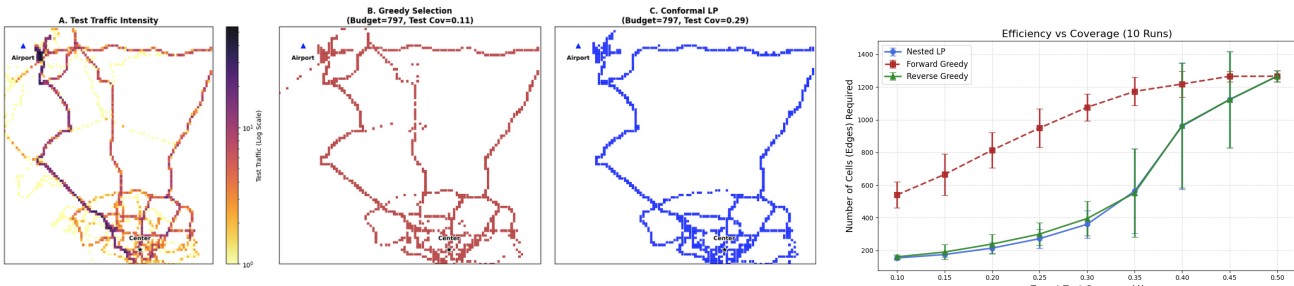

*(a)* **(Left)** Heatmap of test traffic. **(Center)** and **(Right)** Edge selection for Forward Greedy and LP respectively with the same edge budget of 797 edges.

*(b)* Greedy and LP compression over 10 runs.

*Figure 2.* Real-world validation on Porto Taxi data (Airport → Center).

Sampling reduces to picking an activity $v \in G_r$ with probability proportional to $N(r+1, K - w_v)$ and recursing, which allows efficient uniform sampling of itineraries from $S_{d^*}(A^*)$.

## F. Proof of Theorem 4.4

*Proof.* We will use the framework of (Gao et al., 2025b) to get the conformal coverage guarantee. In fact, the approach of starting with a set with a size approximation guarantee to a conformity score function using a nested family is almost identical to Section 7 of (Gao et al., 2025a). So we focus on how to adapt it to our setting. From the uniform convergence result in Lemma E.1, we can run the LP rounding algorithm on a sample of size $T/2 = \Omega(|V|/\gamma^2)$ to obtain a set $K$, that using Theorem 4.1, already satisfies Guarantee 2 (Compression) of Theorem 4.4.

Now, to construct a conformal predictor, it suffices to give a *conformity score*. The procedure in Section 4.3 constructs a *nested set system.* We can use this set system to construct a conformity score, see e.g. Equation (10) and Assumption 2.4 in (Gao et al., 2025b), which implies a conformal predictor via split conformal prediction. They have the property that they will only output sets $S$ from the input nested set system $S_1, \ldots, S_k$. Hence split conformal prediction ensures that as long as $Y_1, \ldots, Y_{T+1}$ are exchangeable we have $\mathbb{P}(Y_{T+1} \in \hat{K}) \geq \phi$.

Suppose $Y_1, \ldots, Y_{T+1}$ are drawn i.i.d. and $T = \Omega(|V|/\gamma^2)$, then since $K$ is in our nested set system, and Theorem 4.1 guarantees that $K$ achieves coverage $\geq \phi$, the conformal predictor will not output any set in the system larger than $K$. This completes the proof. $\square$

## G. Additional Experiments

### G.1. Real-World Traffic Experiment

We complement the experiment in Section 5 with real-world trajectory data. We use the Porto Taxi Trajectory Dataset (Moreira-Matias et al., 2013). On this dataset, the number of samples is small enough that the conformal guarantee $\phi$ differs significantly on the held-out set compared to the compression parameter $\tau$, so that we are not in the "uniform convergence" regime. We show that the nested subgraph approach in Section 4.3 achieves robust compression in this regime (in addition to providing a worst-case theoretical compression guarantee).

The dataset consists of 1.7 million taxi trajectories in Porto, Portugal. We define a bounding box around the Greater Porto area and discretize this region into a $100 \times 100$ grid. Each valid grid cell ($\approx 100m \times 100m$) containing at least one GPS reading constitutes a vertex in our hypergraph. Each taxi trip is mapped to the set of unique grid cells it traverses, forming a hyperedge.

We filter on trips from the airport to the city center, resulting in around 300 trips. Using $k$-means clustering on trajectory midpoints, we identify two distinct route modes naturally taken by drivers: The trips that take highways that circle around the city and the trips that cut diagonally (likely taking city streets). We construct a dataset with 40% of the former and 60% of the latter trajectories. We split the data randomly into 100 train and 100 disjoint test routes. We use the training data to find the LP and greedy solutions, and use the test data to evaluate coverage.

In Figure 2(b), we note that the LP and the reverse greedy algorithm achieve comparable and non-trivial compression for

$\phi \leq 0.4$ relative to the forward greedy baseline. Note in this case that the number of paths is not large enough to ensure a "uniform convergence" type behavior in the samples (as in Theorem 4.4), so that the conformal guarantee even with no compression is $\phi \leq 0.5$. Nevertheless, we observe that our algorithm provides robust compression in this regime. The edge selections when the LP and Forward Greedy procedures are allowed 797 edges (corresponding to $\tau = 0.8$ for the LP) are shown in Figure 2(a). In this setting, greedy prioritizes an additional highway to the airport, while the LP prioritizes the city center.

One interesting observation is that reverse greedy and LP achieve comparable performance on this instance, suggesting that the empirical route distribution does not exhibit strong compressible structure beyond what is captured by frequency-based heuristics. As mentioned before, our goal in this experiment is to demonstrate that the nested compression approach provides a strong compression guarantee even in the sparse sample regime where the conformal guarantee differs significantly from the $\tau$ used on the training samples. Furthermore, the LP rounding algorithm always provides a compression guarantee in the sense of Theorem 4.1, and as the example below shows, reverse greedy (or forward greedy) cannot provide such a guarantee in general.

**Example G.1.** *Fix some relaxation parameter $\kappa > 0$ as in Theorem 4.1. Choose some $\epsilon > 0$ so that $1 + \kappa \leq \frac{1-\epsilon}{\epsilon}$. There is a path $P$ with $a$ edges between $s$ and $t$ with $w_P = \epsilon$ fraction of traffic. In addition, there is a set $S$ of $b \ll a$ parallel edges between $s$ and $t$ each with $w = \frac{1-\epsilon}{b}$ fraction of traffic. Suppose the goal is to cover $\tau = 1 - \epsilon$ fraction of traffic. Then the optimal solution deletes $P$ and incurs cost $b$ in terms of number of edges. Reverse greedy deletes edges from $S$ until the deleted coverage is $(1 + \kappa) \cdot \epsilon$ and incur edge cost at least $a \gg b$, so that no worst-case compression guarantee analogous to Theorem 4.1 is possible.*

### G.2. Trip Planning

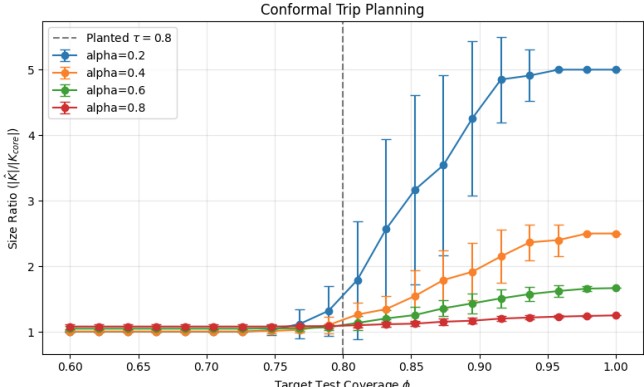

*Figure 3.* Trip Planning simulation with planted core probability $\tau = 0.8$. The x-axis is the conformal guarantee $\phi$, and the y-axis represents the ratio of selected vertices to the optimal core size.

We next investigate a structured recommendation scenario where the goal is to recommend a bundle of activities (e.g., a vacation itinerary). We simulate a domain with $R = 5$ distinct types of activities (e.g., dining, museums, hiking). For each type $r$, there is a set $A_r$ of available activities with $|A_r| = 10$. Within each type, we designate a "core" subset $A_r^*$ representing high-probability options, where $|A_r^*| = \alpha |A_r|$ for a density parameter $\alpha \in [0, 1]$. The total vertex set is the union of all $A_r$. We generate hyperedges (itineraries) by selecting exactly one activity from each type. The generation process follows a planted model: For each type $r$ independently, the activity is drawn uniformly from the core $A_r^*$ with probability $p$, and uniformly from the complement $A_r \setminus A_r^*$ with probability $1 - p$. The parameter $p$ is calibrated such that $p^R = \tau$, meaning that a "pure core" itinerary (composed entirely of core activities) occurs with probability $\tau$. In this setting, the ideal conformal set for mass $\tau$ is exactly the union of the core subsets, having a total size of $\sum |A_r^*| = \alpha \cdot |A_r| \cdot R$.

We fix the planted core probability $\tau = 0.8$ and vary the core density $\alpha \in \{0.2, 0.4, 0.6, 0.8\}$. As before we generate 100 training samples $\{Y_j\}_{j=1}^{T/2}$ and run the algorithm in Section 4.2 to obtain a nested sequence of subgraphs. For each conformal parameter $\phi$, we find the smallest subgraph achieving coverage $\phi$ on 100 held out samples $\{Y_j\}_{j=T/2+1}^{T}$. (In other words, $T = 200$ in the algorithm in Section 4.3.) We plot the ratio of the number of vertices in the subgraph to the size of the ground-truth core, as a function of $\phi$ in Figure 3. A ratio of 1.0 implies perfect recovery of the core.

The results highlight two key findings. First, for all values of $\alpha$, the ratio remains strictly at $1.0$ as long as the target coverage $\phi$ is at most $0.75$, close to the true core probability $\tau$. This confirms that the LP rounding scheme correctly prioritizes the high-density core structure. Second, we observe a sharp "knee" in the performance curves around $\phi = 0.8$. To satisfy a coverage request $\phi > 0.8$, the algorithm is forced to include hyperedges that contain non-core activities. Capturing this marginal probability mass requires adding a large number of new vertices, causing the ratio to rise sharply.

