# OpenReview forum: "Compact Conformal Subgraphs"
_ICML.cc/2026/Conference — ICML 2026 regular_

### Official Review · Reviewer_1srp · 2026-02-23

**Soundness:** 3
**Presentation:** 3
**Significance:** 3
**Originality:** 3
**Overall Recommendation:** 4
**Confidence:** 3

**Summary:**

Rigorous, distribution-free uncertainty guarantees are provided by conformal prediction, but in structured domains such as routing, planning, or sequential recommendation, prediction sets are often produced that are prohibitively large. A framework for constructing compact subgraphs that preserve statistical validity while reducing structural complexity is introduced, referred to as graph-based conformal compression. Compression is formulated as the problem of selecting the smallest subgraph that captures a prescribed fraction of the probability mass. This formulation is reduced to a weighted variant of the densest-k-subgraph problem in hypergraphs, in the regime where a large fraction of edges is contained in the subgraph.  A key monotonicity property is proven to be satisfied by the proposed relaxations, and this property is derived from a connection to parametric minimum cuts. The nestedness required for valid conformal guarantees is ensured by this monotonicity property. A bridge between conformal prediction and combinatorial graph compression is established through this connection, and rigorous guarantees on both statistical validity and compression size are provided.
Finally, the proposed algorithmic approach is validated through simulations in trip planning and navigation tasks, and comparisons with natural baseline methods are conducted.

**Compliance With Llm Reviewing Policy:**

Affirmed.

**Final Justification:**

The detailed rebuttal by the authors addressed my concerns through appropriate clarifications and better highlighting of the novelty.

**Key Questions For Authors:**

Please refer to weaknesses.

**Strengths And Weaknesses:**

**Strengths**

S1: Interesting links between conformal prediction and dense subgraph detection.

S2: The paper benefits from clear presentation and organization (in most parts).

**Weaknesses**

W1: The motivation behind compressing conformal sets is not always clear. For example, in the setting of Section 2.1, in some cases the conformal sets can be intrinsically large, because the model is unreliable and its output $A$ deviates substantially from the ground-truth $B$ in terms of the non-conformity score $f$. What is the motivation of applying compression in such cases, and what are the implications for the authors' results? I think it makes more sense to tryout compression in the context of conditional coverage [R1]?

[R1]: V. Vovk, Conditional validity of inductive conformal predictors, in Proceedings of the Asian Conference on Machine Learning, vol. 25, 2012, pp. 475490.

W2: The proof of Lemma D.1 needs greater elaboration. It is not clear which result of (Charikar 2000) is being referenced, which applies for the case of edges, instead of hyper-edges. I think that an implicit modelling assumption is being made under the hood that applies a clique-expansion of the hyperedge, but these technical details have been swept under the rug. For completeness, I request the authors to furnish a self-contained proof.

W3: There are several extensions of the work of (Goldberg 1984, Gallo 1989, Charikar 2000) to extract dense subhypergraphs, which also involve solving a min-cut problem on a flow network. It is not clear what to what extent the one proposed by the authors in the proof of Lemma D.2 is different, or whether their result is subsumed by the pre-existing literature (see [R2], [R3], [R4]).

[R2]: Tsourakakis, Charalampos. The k-clique densest subgraph problem. In Proceedings of the 24th international conference on world wide web, pp. 1122-1132. 2015.

[R3]: Chekuri, Chandra, Kent Quanrud, and Manuel R. Torres. Densest subgraph: Supermodularity, iterative peeling, and flow. In Proceedings of the 2022 Annual ACM-SIAM Symposium on Discrete Algorithms (SODA), pp. 1531-1555. Society for Industrial and Applied Mathematics, 2022.

[R4]: Huang, Yufan, David F. Gleich, and Nate Veldt. Densest subhypergraph: Negative supermodular functions and strongly localized methods. In Proceedings of the ACM Web Conference 2024, pp. 881-892. 2024.

W4: The complexity incurred in solving the LP (1) is not discussed. Since the hyper-edges do not need to be uniform in size, the number of constraints appearing in (1) can be large. The authors tackle this problem by showing that a polynomial size random sample is sufficient, but a concrete time complexity bound would be helpful.

W5: The bi-criteria approximation guarantees of the relax and round strategy does not seem to be affected by whether the number of edges in the graph is linear or sublinear? This assumption is not invoked in the proof of Theorem 4.1?

W6: In Section 4, the coverage parameter is used in several forms ranging from $\tau$, $\epsilon$ and $\kappa$. It would be helpful if all statements of the main results were expressed in terms of a single quantity.

---

> ### Author Rebuttal · Authors · 2026-03-27
>
> We thank the reviewer for the careful reading and detailed feedback.
>
> W1: The motivation for conformal prediction comes from interpretability. Presenting an end-user with a raw, unstructured set of thousands of potential trajectories is impractical due to cognitive overload. Unlike a set of paths (which could be exponential in size), our conformal set (a subgraph) is a tractable object even when the model is unreliable.  Our framework summarizes this into a compact, easily interpretable subgraph (e.g., highlighting a few main traffic arteries on a map). If the base model is unreliable, the optimal conformal subgraph will naturally be larger. However, our theoretical results still hold,  and the subgraph returned could still be much smaller than outputting all paths.
>
> We note that conformal prediction has typically focused on conformal validity, with very  few recent results focusing on provable (as opposed to heuristic) efficiency of the solution (in our case, the size-optimality of the compressed graph). However, both provable validity and efficiency are a requirement, since validity is often trivial without efficiency (e.g., just output the entire graph). We view one of our main conceptual contributions as being the connection between conformal prediction, graph compression and approximation algorithms. Conversely, this framework provides a novel systems motivation for conformal prediction itself: by using past data to identify a high-probability, compressed subgraph, we safely prune the network and reduce the search space for routing, planning, and optimization algorithms.
>
> Connection to Conditional Coverage [R1]: We agree that conditional coverage is practically desirable. As established in the literature (including by [R1]), this is generally impossible to achieve without making strong assumptions. However, our framework is orthogonal to, and compatible with conditional coverage. For instance, if one partitions the calibration data into difficulty categories, our algorithm can simply be applied within each category, satisfying category-wise conditional coverage. We will add a discussion on this.
>
> W2: We will provide a self-contained proof in the revision and clarify that we do not rely on a clique-expansion of the hyperedges. The result from Charikar applies directly to our hypergraph setting without modification. Pick a threshold $\alpha \in[0, 1]$ uniformly at random, and set $x_v = 1$ if $x_v \ge \alpha$ (and $0$ otherwise). Setting the vertices this way ensures the constraints are preserved, and the Lagrangian objective is preserved in expectation.
>
> W3: We thank the reviewer for the pointers. The Lagrangian for DkS coincides with that for the densest ratio subgraph, so it is unsurprising the natural flow network construction in Lemma D.2 is not fundamentally new and is indeed present in [R3]  for the densest ratio subhypergraph problem. (This is also the reason why we did not provide a proof for Lemma D.1.) We will add these citations and clarify this context in the revision.
>
> However, our problem is DkS and not the ratio subgraph, and the primary theoretical contribution in this section is not the flow network itself, but rather proving that our approximation algorithm for DkS satisfies the monotonicity requirement of conformal prediction (Lemmas D.3 and D.4).  This chain of monotonicity is the prerequisite for valid conformal guarantees. We will highlight our exact contribution better.
>
> It is quite surprising to us that the sequence of graphs found by simply optimizing the Lagrangian for varying $\lambda$ is not just monotone, but is also an approximation algorithm for the “miscoverage” version of DkS. This connection and reinterpretation of an existing algorithm is a priori not obvious. We will expand Section 4.2 to explicitly detail this and make our novel contribution clearer.
>
> W4: Corollary 4.3 shows the time complexity of finding all the relevant LP solutions. The running time there is borrowed from [Gallo et al.].
>
> W5: The proof of Theorem 4.1 does not invoke an assumption about the number of edges being linear vs. sublinear. The inequality $W_{\text{lost}} \le (1+\kappa)\epsilon W$ holds unconditionally for any graph.
>
> However, this bound is only meaningful when the target subgraph retains a large fraction of edges (i.e., $\epsilon$ is a small constant). In the classical Densest $k$-Subgraph (DkS) regime, the target subgraph contains a sublinear fraction of the edges, meaning $\epsilon \to 1$. Then our bound becomes $W_{\text{lost}} \le (1+\kappa)W$, which is vacuous. Therefore, while the proof requires no assumptions, the utility of the bicriteria approximation is mainly in the high-coverage (linear) regime, which bypasses classical DkS hardness barriers. This (along with monotonicity) is our main algorithmic contribution, and we will clarify this nuance in the text.
>
> W6: We will clarify the relationship $\tau = 1 - \epsilon$ in the text to avoid confusion.

---

> > ### Author Rebuttal · Reviewer_1srp · 2026-04-02
> >
> > I thank the authors for their detailed rebuttal. I have updated my score.

---

### Official Review · Reviewer_noy8 · 2026-03-12

**Soundness:** 3
**Presentation:** 3
**Significance:** 3
**Originality:** 3
**Overall Recommendation:** 4
**Confidence:** 3

**Summary:**

This paper studies conformal prediction in the setting where the output has a graph structure, where standard conformal prediction techniques may provide a very large prediction set. The authors propose a new conformal prediction method based on graph compression, and provide guarantees on statistical validity, and compression or size.

**Compliance With Llm Reviewing Policy:**

Affirmed.

**Final Justification:**

This is overall a good paper. However as mentioned in the original review, the paper could benefit from more extensive empirical studies. As such, I keep my score at 4.

**Key Questions For Authors:**

Please see weaknesses

**Limitations:**

Yes

**Strengths And Weaknesses:**

Strengths:
1. The connection between conformal prediction and densest subset problem seems interesting.
2. The authors provide guarantees on the statistical validity (coverage) as well as compression (size).
3. The observation that the densest subgraph problem in the conformal compression regime can be approximated efficiently is an interesting insight.

Weaknesses:
1. In some applications (e.g., route planning), users may be more interested in obtaining prediction sets consisting of possible routes. Compressing these routes into a subgraph may produce an output that is no longer a route, which may reduce interpretability and practical utility.
2. Empirical evaluation is limited. In the synthetic experiment, the proposed method performs similarly to the baseline when coverage level is 0.9 and 1. In the real data experiment, the proposed method performs similarly to reverse greedy method.

---

> ### Author Rebuttal · Authors · 2026-03-27
>
> We thank the reviewer for the comments.
>
> We appreciate the reviewer’s comments about the importance of obtaining guarantees on size and efficiency in addition to validity, and the new connection to variants of densest subgraph problems. We are hopeful that this new conceptual connection to graph compression and approximation algorithms will lead to new methods and techniques for achieving size or efficiency guarantees for other conformal prediction problems in discrete or combinatorial settings. Conversely, this framework provides a novel systems motivation for conformal prediction itself: by using past data to identify a high-probability, compressed subgraph, we safely prune the network and drastically reduce the search space for downstream routing, planning, and optimization algorithms.
>
> Hence we view our paper as being primarily a theoretical paper, with the main contributions being conceptual or theoretical (in establishing provable validity and efficiency guarantees for conformal compression). We view our experiments as a proof of concept that the proposed method is likely to work well in practice.
>
> (W1) We agree that if a model is accurate, outputting a single (or small set of) recommended route(s) is ideal. However, in route planning, a standard conformal prediction set might contain hundreds or thousands of distinct overlapping paths (the number of short s-t paths could be exponential in size even in simple series-parallel graphs). Visually presenting a "bag of thousands of paths" to an end-user can be uninterpretable due to cognitive overload.  Our subgraph compression is compact and acts as a visual summary, so that on a map, the user instantly sees the geographic envelope that contains the ground-truth route with $\ge \phi$ probability. Even when the model is unreliable, the conformal set (the subgraph) is a tractable object that is not of exponential size. Furthermore, in applications like our Trip Planning experiment (Section 5.2), the subgraph directly corresponds to a narrowed-down ``menu'' of highly probable activities (e.g., specific museums and restaurants), which is interpretable and directly useful for the end-user. We will add a brief discussion clarifying this practical usage in the revision.
>
> (W2) In the synthetic routing experiment (Figure 1c), the LP solution is significantly more compressed than both greedy baselines for all coverage levels $\phi \le 0.8$, which is a reasonable setting for compressing large, unwieldy sets of paths. The algorithms only converge as $\phi \to 1.0$ because, mathematically, to cover nearly 100\% of the routes, any algorithm is forced to select nearly all the edges in the graph.
>
> Further, we view the fact that the LP matches the best greedy heuristic on the real-world dataset as a positive result: it shows our algorithm is practically competitive with natural heuristics, while providing a theoretical worst-case guarantee that greedy methods lack (see Example G.1).

---

> > ### Author Rebuttal · Reviewer_noy8 · 2026-04-03
> >
> > The authors have addressed most of my questions. I will keep my scores unchanged as a more extensive empirical study would strengthen the paper.

---

### Official Review · Reviewer_Pomy · 2026-03-12

**Soundness:** 3
**Presentation:** 3
**Significance:** 3
**Originality:** 2
**Overall Recommendation:** 4
**Confidence:** 2

**Summary:**

The work introduces the problem of compressing the set in output to conformal prediction models for graph problems. The authors study two different settings Distribution-free and distribution-based. The work designs methods based on $k$-densest sub(hyper)graph to solve the problem of compression. Experiments against greedy algorithms show the effectiveness of the approach.

**Compliance With Llm Reviewing Policy:**

Affirmed.

**Final Justification:**

Most of my concerns have been addressed in the rebuttal by the authors.

**Key Questions For Authors:**

Address the weak points

**Limitations:**

yes

**Strengths And Weaknesses:**

Strengths

S1. The problem of outputting a small conformal prediction set is important and challenging.
S2. The paper is easy to follow.
S3. The application to road networks is interesting.

Weaknesses

W1. The techniques appear incremental.
A. The authors map the problem on a $k$-densest subgraph formulation and rely on textbook LP-rounding to achieve a bicriteria algorithm.
B. The sampling-based algorithm (for when the number of hyperedges is exponential in $n$) has high complexity and can only be applied on weights, making it not useful for practical applications where scalability is an issue.
C. There is no clear computational complexity analysis, and notation hides the exact time-complexity.

W2. There are many assumption that need to be validated.
A. The authors assume exchangeability of data. It is unclear to me under which data such assumption holds. If the assumption does not hold on real data, than most of the guarantees obtained by the authors cannot hold in practice.
B. The writing of the paper is too specific about road planning, do the techniques proposed hold for general applications?
C. There are hidden assumptions in the sampling algorithm.

W3. Experiments are inconclusive.
A. There are no baselines specifically designed to solve the problem addressed by the authors.
B. To my understanding only one real dataset and synthetic datasets are considered.
C. There is no validation of the proposed bounds.

---

> ### Author Rebuttal · Authors · 2026-03-27
>
> Thank you for your comments.
>
> We would like to emphasize that the main contribution of the paper is the conceptual framing, and the theoretical guarantees that we show in this paper.  The conformal prediction literature has predominantly focused on conformal validity, with only a few recent results focusing on provable (as opposed to heuristic) efficiency of the solution (in our case, the size-optimality of the compressed graph). However, both provable validity and efficiency are a requirement, since validity is often trivial without efficiency (e.g., just output the entire graph). The conceptual contribution in this regard is framing it as a graph compression problem, and its connection to classic problems in approximation algorithms. We build on these ideas and also show non-trivial nestedness properties required for use in conformal prediction. Overall, we are able to achieve rigorous guarantees for both validity, and the size (or efficiency) for a simple algorithm. We are aware of few works that show such size guarantees on the conformal set.
>
> We would also like to address and respectfully push back on some of the specific weaknesses mentioned in the review.
>
> Response to W1 (Techniques).
>
> A. Novelty of the LP approach: While the bicriteria LP rounding itself has appeared in other contexts, our primary contribution is bridging this with the statistical requirements of Conformal Prediction. It is non-trivial to prove that the threshold rounding exhibits monotonicity with respect to the coverage target $\tau$ (Lemmas D.3 and D.4). This monotonicity is required for valid conformal calibration. Further, even viewed as a pure approximation algorithms problem, it is quite surprising to us that in the regime where the coverage requirement is large, relaxing the mis-coverage enables a constant approximation to the size of the compressed subgraph via a simple, monotone algorithm – this result is novel as far as we are aware. Please also see response to Reviewer ​​1srp.
>
> B. Sampling complexity and weights: The sampling algorithm (Appendix E.2) runs in $O(d^* \cdot |E|)$ time. It handles integer/discretized distance metrics efficiently. Furthermore, Appendix E.2.2 shows how this DP approach generalizes to other structures, such as hierarchical classification trees.
>
> C. Computational complexity: The exact time-complexity of our algorithm is explicitly stated in Corollary 4.3. Computing the nested sequence of subgraphs takes $\tilde{O}(\gamma (m + n)^2)$ time, where $\gamma$ is the maximum hyperedge size, $n$ is vertices, and $m$ is hyperedges. We will make this more prominent in the main text.
>
> Response to W2 (Assumptions and Scope):
>
> A. Exchangeability: Exchangeability is not a new assumption introduced by our paper; it is the fundamental assumption underlying the field of distribution-free Conformal Prediction (e.g., Vovk et al., 2005). Any data that is drawn i.i.d. from an unknown distribution is exchangeable. Thus, our guarantees hold in the same regimes as all classical conformal prediction methods.
>
> To clarify further, we are not assuming the nodes or edges of the graph are exchangeable. The assumption applies to the data – the pairs of (model prediction, ground-truth trajectory) $(A_t, B_t)$. As long as the users' trips are drawn i.i.d. (or exchangeably) from an underlying distribution of routing behavior, conformal guarantees hold.
>
> B. Generality beyond road planning: Our framework is general to any setting that can be modeled as a hypergraph. We demonstrate this in Section 5.2 (Trip Planning), which models a combinatorial recommendation system. Furthermore, Appendix E.2.2 shows how to formulate hierarchical classification (where the hypergraph is a tree of labels), a setting also considered in Zhang, Li, and Bastani.
>
> C. Hidden assumptions in sampling: Our assumptions are fairly mild and explicitly stated in the first paragraph of Appendix E.2.1 (e.g., allowing walks instead of simple paths). For other domains like hierarchical classification (E.2.2), there are no assumptions.
>
> Response to W3 (Experiments and Baselines):
>
> A. Baselines: Because graph-based conformal compression is a novel problem introduced in this paper, we compared against natural frequency-based Greedy and Reverse-Greedy algorithms, which are the standard heuristics for densest-subgraph-style problems. These serve as strong and natural baselines.
>
> B \& C. Validation of bounds: Our experiments directly validate our theoretical bounds. Figure 2 validates the exact recovery behavior of the LP: the algorithm perfectly recovers the exact ground-truth core structure (ratio of 1.0) until the target coverage $\phi$ exceeds the planted mass $\tau=0.8$. Figure 1(c) empirically validates the compression vs. coverage tradeoff. Finally, Example G.1 provides a theoretical validation of why the LP is necessary: it shows that natural greedy heuristics can fail arbitrarily badly on adversarial topologies.

---

> > ### Author Rebuttal · Reviewer_Pomy · 2026-04-03
> >
> > Thank you for the response. My concerns have been addressed, and I will adjust my review accordingly.

---

### Official Review · Reviewer_89ci · 2026-03-13

**Soundness:** 3
**Presentation:** 3
**Significance:** 3
**Originality:** 3
**Overall Recommendation:** 5
**Confidence:** 3

**Summary:**

Motivated by structured prediction tasks common in the logistics and
recommendation domains, the paper tackles the challenge of building prediction
graphs that are both statistically valid (achieve marginal coverage) and
efficient (as small as possible). Tools from the conformal prediction and
densest subgraph discovery literature are used to achieve that goal, leading to
the graph-based conformal compression framework and specific algorithms. The
method is evaluated on synthetic and real-world datasets in a fixed-context
setting.

**Compliance With Llm Reviewing Policy:**

Affirmed.

**Final Justification:**

The authors addressed my main concerns during the rebuttal period, none of which were major, and I have decided to maintain my recommendation for the paper to be accepted.

**Key Questions For Authors:**

- The monotonicity property $\tau_1 < \tau_2 \implies K_{\tau_1}(A*) \subseteq
K_{\tau_2}(A*)$ in Definition 2.1 resembles the nested-sets assumption
$\lambda_1 < \lambda_2 \implies T_{\lambda_1}(x) \subset
T_{\lambda_2}(x)$ in risk-controlling prediction sets (Equation 1 in
[1]), and I wonder if any of the conformal extensions proposed in [1] could be
useful for graph compression. Are the authors aware of that work and have they
investigated the extensions therein?

[1] Stephen Bates, Anastasios Angelopoulos, Lihua Lei, Jitendra Malik, Michael Jordan (2021). Distribution-free, Risk-controlling Prediction Sets. Journal of the ACM.

- The conformal validity result in Lemma 2.2 (and similarly in Theorem 4.4)
feels a bit coarse, in the sense that it does not distinguish between the
following two scenarios: (i) $K_{\tau*}(A*)$ and $B*$ are disjoint; and (ii)
$K_{\tau*}(A*)$ covers $B*$ for all but one vertex. In practice, (i) could be
much more problematic than (ii). Have you considered allowing for (small)
miscoverage at the instance level? One approach in this direction could be to
use a different loss function, as in the framework of risk-controlling
prediction sets mentioned above.

- Another object of practical interest is conditional coverage. Can anything be said about the proposed method beyond the marginal results presented?

- How reasonable is the exchangeability assumption on graphs? Have you tried to prove results in more general settings?

- Lemma E.1 is presented without an explicit proof, but with reference to a book. Even if standard, it is usually helpful to include the exact results from the book that imply the Lemma.

- Conformal LP can output disconnected graphs, as in Figure 3 in Appendix G.
Does this mean the graph could be made smaller while retaining the exact same
test coverage, simply by removing isolated edges? I also wonder how much test
coverage would improve by adding the few missing edges needed to make the graph
a connected one. For a more principled approach, how feasible do the authors
believe it would be to enforce connectivity during the graph compression step
(for the problems in which connectivity makes sense, such as the routing
example)?

- Could the authors clarify the following sentence from the proof of Lemma 2.2:
"the subgraph $K_1$ is a subset of $S_{d*}$, so it cannot contain $B*$"? It
does not currently parse, as far as I can tell.

**Limitations:**

Yes.

**Strengths And Weaknesses:**

## Strengths

- Original and relevant: the graph-based conformal compression framework introduced in the paper makes a novel use of conformal prediction and combinatorial graph compression that enjoys immediate application in important problems such as route planning and navigation, illustrated in several experiments.

- Theoretical results: the theory developed covers the classical distribution-free setting as well as a setting in which model probabilities are available, and applies under standard exchangeability assumptions for marginal coverage guarantees and slightly stronger iid and sample size assumptions for compression efficiency.

- Clear: the presentation is sufficiently clear and includes discussions on prior work both in the conformal and densest subgraph discovery literatures.

## Weaknesses

- Experiments and baseline: experiments do not cover the distribution-based
setting (Section 2.2) and therefore also do not include comparisons that
would be relevant there, focusing instead exclusively on the distribution-free
fixed-context setting.

- Minor: some papers in the appendix refer to preprints even though they have
already been published (e.g., Zhang et al. (2025) has been published at ICLR
and Taufiq et al. (2022) has been published at NeurIPS); it would be good to
standardize notation (e.g., $\mathbb{P}[\cdot]$ and $\mathbb{P}(\cdot)$ are currently used
interchangeably); no code was provided, so results are not reproducible;
\citep or \parencite are used for all citations but
\citet or \textcite should be used for some; there are
some typos but they do not hinder understanding (e.g., "correspond" should be
"corresponding" in line 78; "frameworks" should be "framework" in line 42; "the
the" should be "the" in line 147, $G$ in line 148 is not defined.)

---

> ### Author Rebuttal · Authors · 2026-03-27
>
> Thank you for the detailed feedback, and careful reading. It is quite appreciated! We will address all minor comments. Here are the responses to your questions:
>
> (Minor Weakness) We did provide the Python code in the supplementary material. After the paper is accepted, we will publish the colab folder and put a link to it in the paper.
>
> (Q1) Our monotonicity property (Definition 2.1) is equivalent to the nested-sets assumption required by the RCPS framework. This is a synergistic connection: RCPS provides a statistical framework to control general loss functions (e.g., controlling the expected fraction of missed route segments, or some other measure of deviation) rather than just 0-1 marginal coverage, but it requires a nested sequence of sets as input. This works to our advantage: By feeding our nested conformal subgraphs $\{K_\tau\}$ into the RCPS framework, one could immediately extend our work to control arbitrary user-defined risk metrics over graphs. We are very grateful for this pointer; we will cite Bates et al. (2021) and discuss this extension in Section 2 and the Conclusion as another application of our framework.
>
> (Q2) Lemma 2.2 provides a 0-1 marginal guarantee, which is indeed coarse. This is exactly where the connection to RCPS (Bates et al., 2021) is useful (see previous point), and we will highlight this connection to address this comment.
>
> (Q3)  For conditional coverage, as established in the literature, exact finite-sample coverage is generally impossible to achieve without making strong distributional assumptions. However, our graph compression framework is orthogonal to, and compatible with conditional coverage. For instance, if one partitions the calibration data into difficulty categories to achieve approximate conditional coverage, our algorithm can simply be applied independently within each category. This would yield compressed subgraphs that satisfy category-wise conditional coverage. We also remark that recent work e.g., by (Gao et al. 2025) building on (Chernozhukov et al. 2021) have shown how algorithms for the “unsupervised setting” can be used as a subroutine to achieve conditional coverage along with compression/ size guarantees when given access to the conditional distribution. We will add a discussion and clarify how our compression step acts as a modular add-on.
>
> (Q4) To clarify, we are not assuming the nodes or edges of the graph are exchangeable. The exchangeability assumption applies to the data samples, specifically, the pairs of (model prediction, ground-truth trajectory) $(A_t, B_t)$. As long as the users' trips are drawn i.i.d. (or exchangeably) from an underlying distribution of routing behavior, standard conformal guarantees hold.
>
> (Q5) Lemma E.1 is a standard VC-dimension uniform convergence bound, see for instance "Understanding Machine Learning" by Shalev-Shwartz & Ben-David, 2014, Theorem 6.8, or Devroye and Lugosi, Chapter 4. In the revision, we will include the exact reference.
>
> (Q6) For problems like s–t paths, our approach should not produce redundant structures like isolated edges or disconnected graphs, since it will first choose paths, and then take the union of the resulting edges. So it should already be parsimonious. The visualization in Figure 3 is an artifact of discretization of the edges.
>
> (Q7) We apologize for the confusing phrasing. The sentence means the following: If the ground-truth $B*$ was lost in Stage 1 ($B* \notin S_{d*}$), it means $B*$ utilizes at least one road segment that is not in $S_{d*}(A*)$. Therefore, no subset of $S_{d*}(A*)$ (not even the maximal subset $K_1$) can fully cover $B*$. We will rewrite this sentence for clarity.

---

> > ### Author Rebuttal · Reviewer_89ci · 2026-04-02
> >
> > Thank you for the detailed responses. I am glad to hear that the pointer to RCPS was fruitful and the resulting extension will make it into the paper. A final suggestion I have is to try and improve the visualization in Figure 3, as it does appear that there are multiple isolated edges.

---

### Decision · Program_Chairs · 2026-04-30

**Decision:**

Accept (regular)

**Comment:**

This paper considers a conformal prediction framework where the "regression variable" is related to a graph. An important example is routing, where the idea is to output a "small set" of routes that contains the true route with pre-specified probability. The authors notice that standard methods can achieve this at the cost out outputting sets of paths that are too large.

The paper proposes a way to "compress" the output of conformal prediction on this set. This is achieved in a general abstract framework, where a certain LP relaxation of a natural "hypergraph cover" problem is studied. The authors show that their method achieves good guarantees in theory (in terms of marginal coverage) and provide a small number of experiments with their method, with positive results.

I think the reviewers would agree with me that the theoretical parts of this paper are its greatest strength. The problem statement and the analysis of the LP relaxation are standout points. Questions were asked in the discussion about potential strengthenings (e.g. conditional coverage) and some details of the proof. The answers were satisfactory, and I surmise that we are all satisfied that the paper is correct and contains nice original ideias.

The main weakness of the paper, which explains the "weak accepts" of most reviewers, is that the empirical part is small. This is a legitimate point however, my own evaluation is that this is a minor issue for the first paper on this specific topic. I have therefore decided to propose acceptance; then again, given the reviews, the paper could also be bumped down.